# Computational assessment of the feasibility of protonation-based protein sequencing

**Giles Miclotte**[1], **Koen Martens**[2]*, **Jan Fostier**[1]

**1** IDLab, Ghent University-Imec, Ghent, Belgium, **2** Imec, Leuven, Belgium

* Koen.Martens@imec.be

**Data Availability Statement:** All protein data files are available from the Swiss-Prot database at uniprot.org, specifically it concerns the H. sapiens data found at: https://www.uniprot.org/uniprot/?query=reviewed%3Ayes+AND+organism%3A9606.

## Abstract

Recent advances in DNA sequencing methods revolutionized biology by providing highly accurate reads, with high throughput or high read length. These read data are being used in many biological and medical applications. Modern DNA sequencing methods have no equivalent in protein sequencing, severely limiting the widespread application of protein data. Recently, several optical protein sequencing methods have been proposed that rely on the fluorescent labeling of amino acids. Here, we introduce the reprotonation-deprotonation protein sequencing method. Unlike other methods, this proposed technique relies on the measurement of an electrical signal and requires no fluorescent labeling. In reprotonation-deprotonation protein sequencing, the terminal amino acid is identified through its unique protonation signal, and by repeatedly cleaving the terminal amino acids one-by-one, each amino acid in the peptide is measured. By means of simulations, we show that, given a reference database of known proteins, reprotonation-deprotonation sequencing has the potential to correctly identify proteins in a sample. Our simulations provide target values for the signal-to-noise ratios that sensor devices need to attain in order to detect reprotonation-deprotonation events, as well as suitable pH values and required measurement times per amino acid. For instance, an SNR of 10 is required for a 61.71% proteome recovery rate with 100 ms measurement time per amino acid.

## Introduction

Proteins form the core of many biological processes such as cell metabolism, cell signaling, cell adhesion, immune responses, etc. The absence, presence or modifications of proteins can thus have various biological or clinical implications [1]. Therefore, the accurate identification of proteins present in a sample can provide information on biological systems, complementing genomics and transcriptomics profiling. Specific protein sequencing applications include for example clinical proteomics, microbial analysis, molecular diagnostics for health monitoring, functional proteomics, bioreactor monitoring and organ monitoring [2–7]. The most common genomic sequencing methods yield short and highly accurate reads with a high throughput, but these methods do not easily translate to proteomics. Second generation DNA (and RNA) sequencing methods [8] typically rely on polymerase chain reaction (PCR) to copy DNA molecules, thus forming clusters of identical DNA molecules. When using fluorescent

**Funding:** This work is funded by imec vzw. Giles Miclotte, Koen Martens and Jan Fostier are employees of imec vzw, Belgium; Giles Miclotte and Jan Fostier are employees of Ghent University, Ghent, Belgium. The funder provided support in the form of salaries for authors [GM, KM, JF], but did not have any additional role in the study design, data collection and analysis, decision to publish, or preparation of the manuscript. The specific role of each author is articulated in the "author contributions" section.

**Competing interests:** The authors have the following interests: This work is funded by imec vzw. Giles Miclotte, Koen Martens and Jan Fostier are employees of imec vzw, Belgium; Giles Miclotte and Jan Fostier are employees of Ghent University, Ghent, Belgium. There are no patents, products in development or marketed products to declare. This does not alter the authors' adherence to all the PLOS ONE policies on sharing data and materials.

labels to identify nucleotides, all DNA molecules within a cluster collectively emit a strong fluorescent signal, leading to highly accurate optical methods. For proteins, no similar copying mechanism exists. As such, next-generation protein sequencing will likely be more similar to third generation DNA sequencing [9, 10] (single molecule sequencing—SMS), in which individual DNA molecules are sequenced without amplification. Recent advances in the development of these SMS methods allow for the generation of very long DNA reads with a high throughput. In this respect, contemporary methods for protein sequencing are significantly more limited.

## Protein sequencing methods

Three established protein identification methods can be discerned, each with their own limitations. First, Edman degradation subjects a peptide to consecutive steps of degradation, allowing to break down and identify the sequence one amino acid at a time. However, this method suffers from a very low throughput and only works for short peptides. Second, immunoassays rely on antibodies to identify proteins. However, these assays require the development of specific antibodies for each protein or protein modification of interest. Finally, mass spectrometry-based methods offer high throughput and require no application-specific antibody design. However, due to the high complexity of protein samples, mass spectrometry often fails to correctly resolve all proteins in a sample. Additionally, mass spectrometry requires expensive and large equipment.

As such, a sequencing method that offers high throughput, high adaptability, high accuracy, and a low cost is not yet available. However, in recent years, several novel protein sequencing techniques have been proposed, based on optical imaging of fluorescently labeled molecules. Although promising, these methods are currently not yet well-established.

In the fluorosequencing technique [11], a specific set of amino acids is fluorescently labeled. Through Edman degradation of peptides with a fluorescently labeled reagent, amino acids are removed one by one, and the fluorescence signal is recorded. Removing a fluorescently labeled amino acid changes the overall fluorescence intensity of the molecule, allowing for the identification of that amino acid. The pattern of degradation events combined with changes in fluorescence intensity allows for identification of peptides in a sample, using a reference database. Similarly to the fluorosequencing technique, a set of amino acids is fluorescently labeled for sequencing based on fingerprinting [12]. However, the protein is not digested into peptides nor degraded. Instead, the polypeptide is guided through a Caseinolytic protease X pore protein. This allows for the sequencing of much longer polypeptides than is possible using Edman degradation. In this method a small subset of amino acids is labeled, for example cysteine and lysine. The measured signal is then a list of occurrences of these amino acids, with their relative distances. This allows for the identification of the polypeptides in a sample using a reference database. In a third method, N-terminal amino acid binding proteins (NAABs) bind to a terminal amino acid of a peptide. Fluorescent labels on the NAABs can then be used to determine binding events and their duration. In this method [13, 14] several (17) NAABs are used consecutively, with each NAAB having different binding affinities for the different terminal amino acids. In this way, two types of information are obtained: (1) which NAABs bind to the terminal amino acid, and (2) how long they were bound. After obtaining this information for an entire set of NAABs, the terminal amino acid is removed, for example by Edman degradation. This information can then again be used to identify the peptide using a reference database. Finally, protein sequencing through peptidase degradation subjects the peptide to consecutive degradation events, while measuring the time it takes for the peptidase to degrade each amino

acid. The degradation happens through the binding and cleaving of a fluorescently labeled peptidase to the N-terminal amino acid of the peptide [15].

## Protonation-based protein sequencing

Here, we propose and computationally assess protein sequencing through reprotonation-deprotonation. In this method, we observe the repeated binding and release of a proton in either the N-terminal amine (-NH$_2$) or C-terminal carboxyl (-COOH) group of the peptide.

- N-terminal amine: -$^+$NH$_3$ $\rightleftharpoons$ -NH$_2$ + H$^+$

- C-terminal carboxyl: -COOH $\rightleftharpoons$ -COO$^-$ + H$^+$

These reaction rates depend on the pH of the solution, and on the unique pKa value of the terminal amino acid. Table 1 provides a list of the pKa values for each of the 20 amino acids. In other words, measuring the average time that the amino acid spends in a certain protonation state allows for its identification.

Fig 1 provides an overview of the reprotonation-deprotonation sequencing method we envision. Proteins are denatured and digested into peptides through the use of an endopeptidase. To obtain N-terminal peptides of proteins only, the protease cleavage of the protein sample is combined with an N-terminal peptide separation chromatography technique [17] such as for example COFRADIC [18]. Chromatographically separating the signature N-terminal peptides from the protease-digested protein sample allows to significantly reduce the peptide sample complexity while still allowing identification of the original proteins. This technique is applied in mass spectrometry-based proteomics [17]. N-terminal peptides are anchored to a substrate, by either one of their terminal amino acids or by a cysteine. In order to obtain approximately a single molecule on a single sensor device a Poissonian single molecule delivery technique is used [19]. Such technique leads to a random distribution of the amount of

**Table 1. The pKa values for the amine groups, carboxyl groups, and side chains that are used in these simulations for the different amino acids [16].**

| 1-letter code | amino acid | carboxyl pKa | amine pKa | sidechain pKa |
|---|---|---|---|---|
| A | Alanine | 2.35 | 9.87 | |
| C | Cysteine | 2.05 | 10.25 | 8.00 |
| D | Aspartic Acid | 2.10 | 9.82 | 3.86 |
| E | Glutamic Acid | 2.10 | 9.47 | 4.07 |
| F | Phenylalanine | 2.58 | 9.24 | |
| G | Glycine | 2.35 | 9.78 | |
| H | Histidine | 1.77 | 9.18 | 6.10 |
| I | Isoleucine | 2.32 | 9.76 | |
| K | Lysine | 2.18 | 8.95 | 10.53 |
| L | Leucine | 2.33 | 9.74 | |
| M | Methionine | 2.28 | 9.21 | |
| N | Asparagine | 2.02 | 8.80 | |
| P | Proline | 2.00 | 10.60 | |
| Q | Glutamine | 2.17 | 9.13 | |
| R | Arginine | 2.01 | 9.04 | 12.48 |
| S | Serine | 2.21 | 9.15 | |
| T | Threonine | 2.09 | 9.10 | |
| V | Valine | 2.29 | 9.72 | |
| W | Tryptophan | 2.38 | 9.39 | |
| Y | Tyrosine | 2.20 | 9.11 | 10.07 |

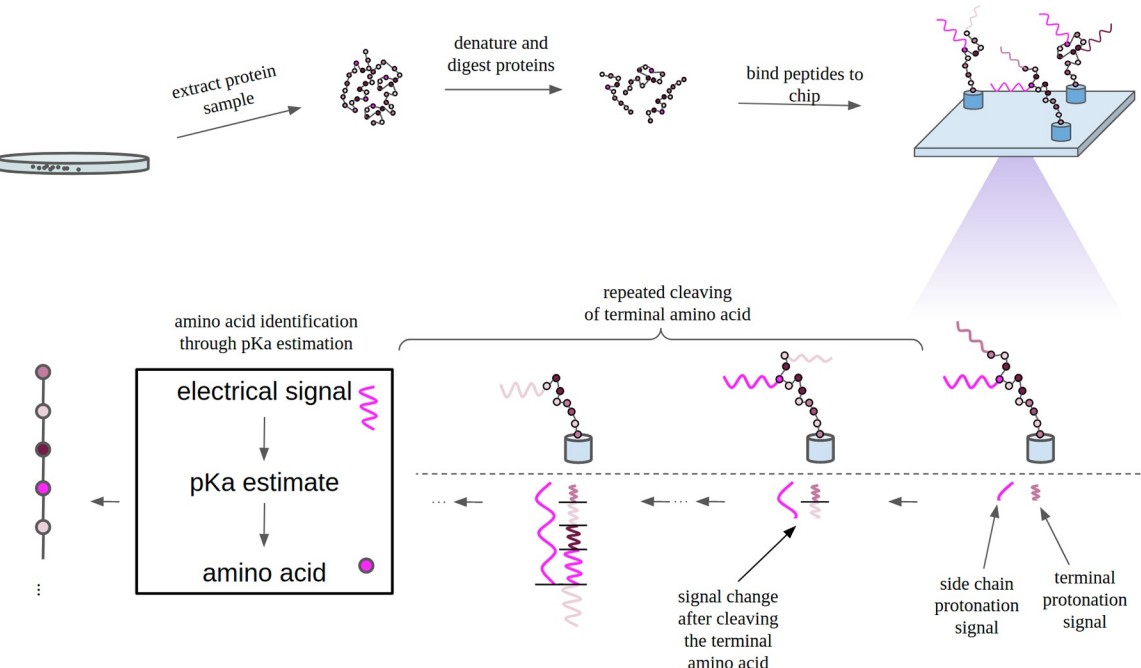

**Fig 1. Protein sequencing based on reprotonation-deprotonation detection.** Proteins are denatured and digested into peptides through the use of an endopeptidase. These peptides are anchored to a substrate, by either one of their terminal amino acids or by a cysteine. Changes in electrical charge due to the reprotonation-deprotonation of the free terminal amino acid are measured at this anchor point. These measurements provide information on the fraction of time the free terminal amino acid spends in a protonated state. In turn, this allows to determine the pKa value and hence identifies the amino acid. By repeatedly cleaving the terminal amino acid, each amino acid in the peptide can be identified in this way.

macromolecules per sensor limited by Poissonian statistics with a theoretical maximum single-molecule occupancy of 37%. A higher single molecule occupation can be obtained for an optimized delivery technique [20]. Molecules can be bound covalently to a surface for instance with Cu-free strain-promoted azide alkyne click chemistry (SPAAC).

Changes in electrical charge due to the reprotonation-deprotonation of the free terminal amino acid are measured. These measurements provide information on the fraction of time the free terminal amino acid spends in a protonated state. In turn, this allows to determine the pKa value and hence identifies the amino acid. Using peptidase degradation or Edman degradation, the free terminal amino acid is removed and the subsequent amino acid in the peptide can be identified. Edman degradation makes use of a harsh acid catalysis step. Chip making materials such as carbon nanotubes and silicon oxide are chemically resistant. An enzyme Edmanase has been proposed [21] to carry out Edman degradation without the harsh acid catalysis step which improves the compatibility with low adsorption detection surfaces.

The electrical signal resulting from reprotonation-deprotonation events is weak and depends on the length of the peptide under consideration, due to shielding. Additionally, the time in between successive reprotonation-deprotonation events is in the order of microseconds or shorter. However, it is possible to observe these events for an arbitrary amount of time. In accordance with the central limit theorem, these repeated measurements of the reprotonation-deprotonation events at a single proton binding site allow for a highly accurate determination of the *average* duration of protonation states, and hence pKa values. These repeated observations offer a major accuracy advantage compared to methods where only a single observation is made for each terminal amino acid. Cleaving the free terminal amino acid can

be achieved using Edman degradation or a peptidase. Edman degradation allows for a precise control over the measurement times, whereas in the peptidase method this happens at random times and the average time depends on the peptidase concentration.

The N-terminal amine group and the C-terminal carboxyl group of a peptide are not the only sources of reprotonation-deprotonation events in a peptide. Several amino acids (cysteine, aspartic acid, glutamic acid, histidine, lysine, arginine and tyrosine) have side chains that can also bind and release a proton (Table 1). These side chain reprotonation-deprotonation events result in additional signals that are superposed on the signal of interest. The side chain pKa values are all significantly higher than the terminal carboxyl group pKa values, but some of the side chain pKa values are similar to the terminal amine group pKa values. Therefore, in the remainder of this work, we identify amino acids through their carboxyl group pKa values, as this reduces the impact of side chain reprotonation-deprotonation events on the signal.

### Hardware to detect reprotonation-deprotonation

Realizing a sensor that can monitor the protonation state of a single molecule is an outstanding challenge. Xiao et al. report on a conductance titration measurement of a single peptide [22]. Their pH-dependent measurement reflects the effect of reprotonation-deprotonation of the amine and carboxyl groups of a single peptide. One of the relatively more established options to detect a single elementary charge of a molecule is the electrolytically gated Field Effect Transistor (FET). In such a device the gate electrode consists of an electrolyte and the ions of the electrolyte charge the gate electrode. The gate electrode bias controls the electrical current in the semiconductor channel. This channel is also sensitive to charges of molecules brought closely to the interface of the channel with the liquid electrolyte. The detection of a single molecule with a FET has been demonstrated by making use of Carbon Nanotube Field Effect Transistors (CNTFETs) [23–27]. By recombinantly modifying two amino acid residues in the protein lysozyme, Choi et al. produced protein variants that differ in charge by elementary units N = -2, -1, 0, 1, and 2. Carbon nanotube FETs were able to distinguish these elementary charge differences, indicating that carbon nanotubes have the potential to monitor the protonation-deprotonation of amino acids [24].

The functioning of an electrical sensor for reprotonation-deprotonation is illustrated in Fig 2. Due to screening by the electrolyte, the signal generated by a charge near the FET channel's surface is expected to decrease exponentially with its distance from the channel's surface. Such exponential dependence has been observed for DNA hybridization [28]. The reprotonation-deprotonation signal of the terminal amino acid may be recorded by making use of a nanoscale electrolytically gated field effect transistor with a peptide closely bound to the FET's channel. In this work, the critical requirements for such a sensor device to allow our protein sequencing method are deduced.

### Results and discussion

Through computer simulations, the feasibility of reprotonation-deprotonation sequencing is explored. For these simulations the following scenario is considered. A mixture of proteins is digested into peptides using an endopeptidase. A single peptide is bound onto a sensor device. As argued before, we opt to bind peptides with their N-terminal amino acid to the anchor point to reduce the impact of side chain protonation events. This leaves the carboxyl group of the C-terminal amino acid free for reprotonation-deprotonation events. Alternatively, the peptides can be anchored with cysteine binding, but this method has not been explored here. The peptides on the substrate are suspended in a solution with known pH and the voltage is sampled at the anchor point at a particular frequency. In this work, we measure at 10 MHz.

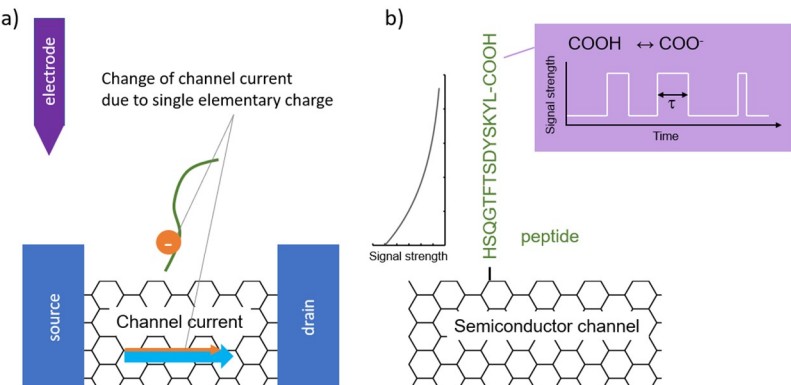

**Fig 2. The functioning of an electrical sensor for reprotonation-deprotonation.** (a) Illustration of the modulation of the channel current by an elementary charge in an electrolytically gated FET. (b)Illustration of the reading out of reprotonation-deprotonation behavior of a peptide by a field effect transistor device. Here, a carbon nanotube device example is given. Signal strength decreases exponentially with distance from the channel's surface. The pKa can be deduced from the observed average on-time.

A Python simulator was implemented to generate a signal from which amino acids are then identified. The signal consists of a high voltage when a proton is bound (on-state) and a low voltage otherwise (off-state) (see Fig 3). In particular, the change in voltage in the underlying signal is proportional to the change in charge at the protonation site. For the carboxyl group, the high voltage state is the neutral state (-COOH), while the low voltage state is the negatively charged state (-COO⁻), i.e. we consider "the voltage difference between the on-state and the off-state." For the amine group the model is straightforward: the high voltage state is the positively charged state (-⁺NH₃), and the low voltage state is the neutral state (-NH₂). As such, deprotonation is the transition from on-state to off-state, and reprotonation is the transition from off-state to on-state. In the case of a carboxyl group this corresponds with a negative charge for the off-state and a neutral charge for the on-state, whereas for the amine group this corresponds with a neutral charge for the off-state and a positive charge for the on-state. The transition rates between these states depend on the terminal amino acid's pKa value, the pH value of the solution, and the times between transitions follow an exponential distribution. To generate realistic signals, we superimpose 1/f noise with a user-defined signal-to-noise ratio. Additionally, side chain contributions are added to the overall signal. Finally, we also take into account the shielding, which depends on the distance between the protonation site and the anchor point of the peptide. This shielding effect increases with the length of the peptide under consideration. Detailed descriptions of the simulator as well as the signal processing algorithm are provided in the Methods section.

## Sequencing isolated amino acids

In this section, the feasibility of *de novo* protein sequencing based on reprotonation-deprotonation sequencing is explored. Single amino acids were sequenced in isolation, i.e. the amino acid signal is not simulated in the context of a peptide. The heatmaps in this section sort amino acids based on their expected on-time, i.e. the expected time that the amino acid is in a protonated state, since amino acids with similar expected on-times are more likely to be confused than amino acids with very different expected on-times. Certain amino acids yield a secondary signal when they have a side chain. The expected on-time takes into account both the main as well as the side chain reprotonation-deprotonation events. As a result, the order of amino acids can vary between heatmaps (depending on the pH of the solution). Each entry

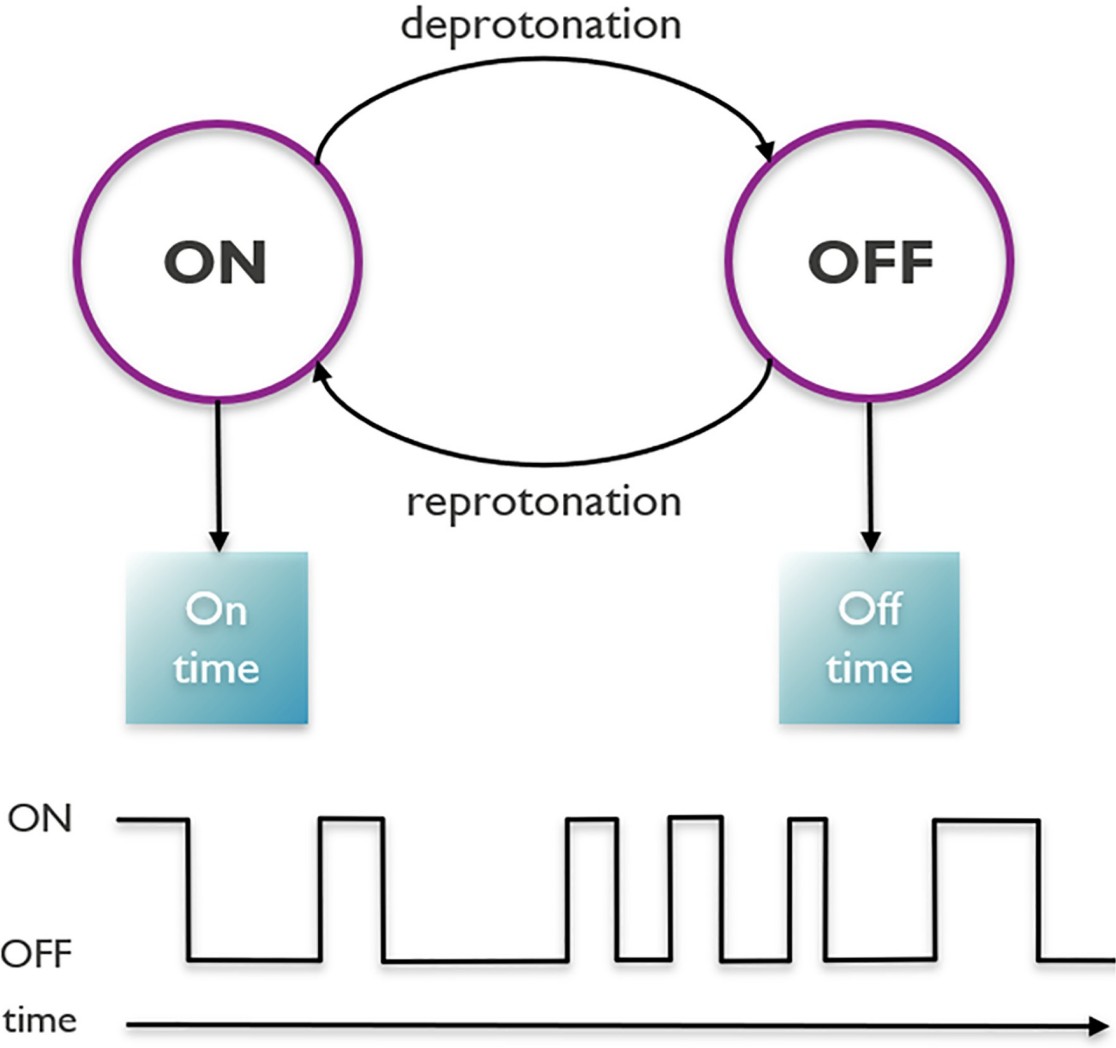

**Fig 3. The simulation model.** There are two states for a given terminal amino acid, and two transitions between these states. A third transition (enzyme cleaving, not shown) leads from the states of the current terminal amino acid to the states of the next, by cleaving the terminal amino acid from the peptide. The amount of time spent in alternating on- and off-states constitutes the measured signal. These on- and off-times are exponentially distributed.

[$i$, $j$] in a heatmap represents the fraction of signals simulated for amino acid $i$ that were identified as amino acid $j$. As such, the values in a row sum to 100%, but the values in a column need not. In this section, we also consider clusters of amino acids, in such a way that amino acids in the same cluster are harder to distinguish with the reprotonation-deprotonation sequencing method, than amino acids in distinct clusters. A cluster is a set of amino acids. In the following each cluster is denoted by a minimal sequence that contains the amino acids of the cluster, for example {C, P, N} will be written as CPN for the sake of brevity.

As a baseline scenario, we consider the simulation of the reprotonation-deprotonation of the acidic carboxyl group at pH 3.0, for a measurement time of 100 ms per amino acid, at a signal-to-noise ratio of 10 (see Fig 4). Certain amino acids, such as phenylalanine, threonine and tryptophan are correctly identified in almost 100% of the cases, whereas it is more difficult to disambiguate between other amino acids. We generated seven clusters (F, CPN, T, QRKYS, MVAGIL, WD, and HE) in such a way that each amino acid has a probability of at least 95% to

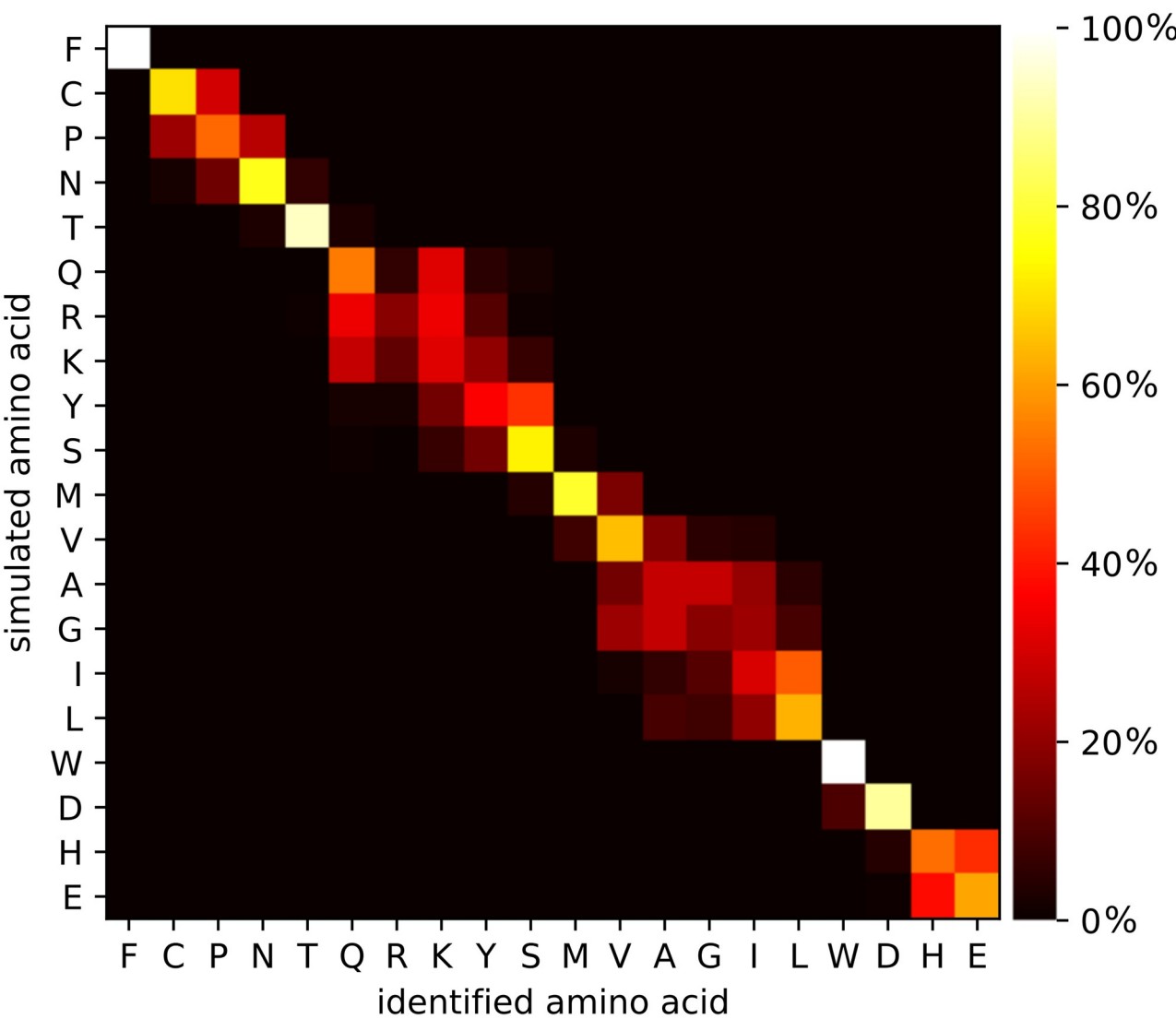

**Fig 4. Heatmap for isolated amino acid identification.** Each entry is the fraction of signals generated for that row's amino acid identified as that column's amino acid. The heatbar shows the interpretation of the colours in the heatmap as percentage of identifications. Sequencing simulations were done for the carboxyl terminal group, at pH 3.0, for 100 ms measurement time per terminal amino acid and an SNR of 10. The average identification accuracy is 59.90%. With 7 clusters (F, CPN, T, QRKYS, MVAGIL, WD, HE), the cluster identification accuracy is >95%.

be identified as a member of the correct cluster. These clusters are already informative: by considering amino acids in the same cluster as indistinguishable and by considering tryptic peptides with maximum length of 50, out of the 19,886 proteins in the H. sapiens database, only 322 proteins (1.6%) can not be uniquely identified with these seven clusters. In contrast, the fluorosequencing approach results in clusters K, DE, Y, W, and FCPNTQRSMVAGILH, which leaves 1,018 proteins (5.1%) not uniquely identifiable. The C-K fingerprinting approach resulting in clusters C, K, and FPNTQRYSMVAGILWDHE, is unable to uniquely identify 9,529 proteins (47.9%).

Heatmaps for lower SNR values 5 and 2 are shown in Fig 5. Unsurprisingly, a lower SNR leads to a lower identification accuracy. At SNR 2, we obtain only three clusters of amino acids (FCPNTQRKYSMVAGIL, WD, HE).

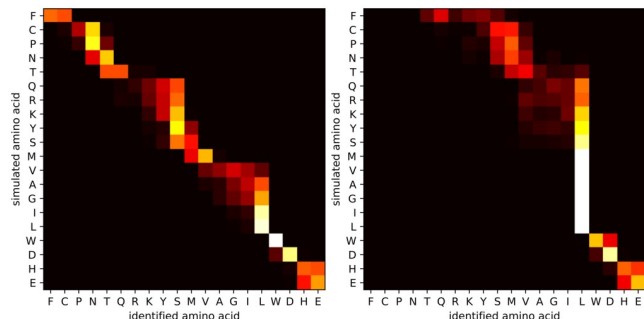

**Fig 5. Heatmaps for isolated amino acid identification for SNR values 5 and 2.** A heatmap for SNR 10 is shown in Fig 4. (left) With an SNR of 5, the average identification accuracy is 33.80%. With 4 clusters (FCPNTQRKYSMV, AGIL, WD, HE) the cluster identification accuracy is >95%. (right) With an SNR of 2, the average identification accuracy is 18.60%. With 3 clusters (FCPNTQRKYSMVAGIL, WD, HE), the cluster identification accuracy is >95%. Sequencing simulations were done for the carboxyl terminal group, at pH 3.0, for 100 ms measurement time per terminal amino acid.

Fig 6 contains heatmaps for measurement times of 1000 ms and 10 ms per amino acid, respectively. These figures confirm that longer measurement times lead to higher accuracies: they allow more samples to be taken from the signal, and more samples lead to better estimates of the average on-time. With a sufficiently long measurement time per amino acid, many small clusters of amino acids can be reliably discerned. In particular, with 10 ms measurement time, the identification accuracy is 32.94% and only 3 clusters can be discerned, while with 1000 ms measurement time, the identification accuracy increases to 79.50%, and the number of discernable clusters increases to 11. This demonstrates that the ability of the method to control the measurement time is a significant advantage.

Heatmaps for different pH values are shown in Fig 7. While the difference in average identification accuracy is rather small, there is a clear effect on the reliable identification of specific amino acids. These heatmaps show how pH can affect the identification rate. At low pH (1.0), amino acids with a low pKa value can be more reliably discerned, whereas at a relatively high pH (4.0), it is more easy to discriminate between amino acids with a high pKa value. In general, the pKa values can be estimated more precisely when the pH value is close to the pKa value and the carboxyl group spends roughly half of the time in either protonated or deprotonated

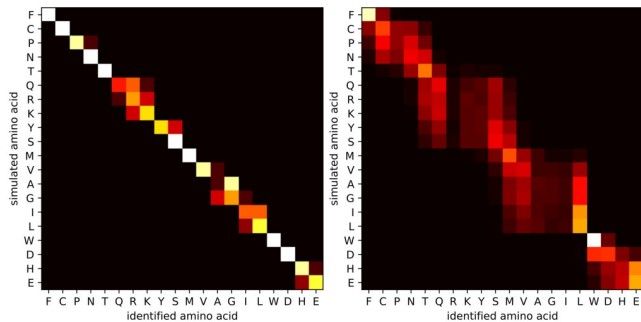

**Fig 6. Heatmaps for isolated amino acid identification for measurement times per amino acid of 1000 ms and 10 ms.** A heatmap for measurement time of 100 ms is shown in Fig 4. (left) With a measurement time of 1000 ms, the average identification accuracy is 79.50%. With 11 clusters (PN, QRK, YS, VAGIL, HE, and 6 singletons) the cluster identification accuracy is >95%. (right) With a measurement time of 10 ms, the average identification accuracy is 32.94%. With 3 clusters (FCPNT, QRKYSMVAGIL, and WDHE), the cluster identification accuracy is >95%. Sequencing simulations were done for the carboxyl terminal group, at pH 3.0, and an SNR of 10.

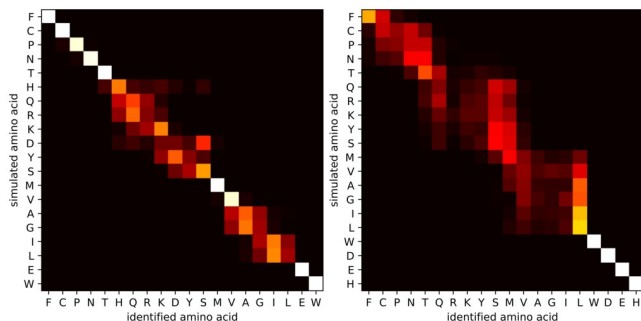

**Fig 7. Heatmaps for isolated amino acid identification at pH 1.0 and 4.0.** A heatmap for pH 3.0 is shown in Fig 4. (left) With pH 1.0, the average identification accuracy is 65.50%. With 10 clusters (HQRKDYS, VAGIL, and 8 singletons) the cluster identification accuracy is >95%. (right) With pH 4.0, the average identification accuracy is 40.80%. With 5 clusters (FCPNTQRKYSMVAGIL, and 4 singletons), the cluster identification accuracy is >95%. Sequencing simulations were done for the carboxyl terminal group, for 100 ms measurement time per terminal amino acid, and an SNR of 10.

state. If the pH is much lower than the pKa value, then the carboxyl group will be in the on-state for the majority of the time. As a result, there will be fewer (but longer) on-time events within a certain fixed measurement time, yielding less accurate estimates for the average on-time and hence less accurate pKa estimates. In addition, one risks missing deprotonation events shorter than the time in between samples. An analogous reasoning can be made when the pH is much higher than the pKa value. In the presence of side chains, the optimal pH might still be different from the average pKa of the amino acids: it can be beneficial to select a very low pH (e.g. 1.0), such that side chains (with pKa at least 3.86) will rarely deprotonate, reducing signal disturbance. On the other hand, these simulations do not take into account the effect that such an acidic environment can have on the peptide and on the device itself.

## Sequencing peptides

In this section, the sequencing of peptides using the reprotonation-deprotonation is explored, in the context of reference-based identification. This means that we select that peptide from a reference database of known peptides, that was most likely to have generated the observed (simulated) signal. We focus on N-terminal tryptic peptides that were derived from human proteins. Electrical signals for peptides were simulated for various parameter settings. Protein identification is simply based on the identification of unique N-terminal peptides. Because not all human proteins have a unique N-terminal peptide, protein identification accuracy will always be lower than the peptide identification accuracy.

When sequencing peptides, the SNR varies between consecutive amino acids in a peptide due to shielding. The longer the distance between the anchoring point of the peptide and the proton binding site, the weaker the signal. As such, the SNR is modeled using two parameters:

1. the initial SNR, i.e. the SNR at the peptide anchoring point, and

2. the shielding factor, i.e. a decreasing function that depends on the peptide length $d$.

The shielding factor is an abstraction that can take into account various physical properties of the system, such as the Debye length, or peptide movement. The shielding factor used in these simulations is of the form $\exp\left(-\chi\sqrt{d}\right)$, where $\chi$ is a positive constant. The square root of the number of amino acids ($\sqrt{d}$) is an approximation of the distance between the proton binding site and the anchor point, assuming a worm-like chain behaviour with a relatively low

persistence length, which is a common model for peptide movement [29]. In order to study the impact of side chain signal interference, we consider both the cases of *bound* and *free* side chains. In the former, the side chains are assumed to be bound to molecules that block side chain reprotonation-deprotonation, thus preventing any signal interference. In the latter, side chains can freely reprotonate-deprotonate.

The baseline simulation uses the following parameters: 100 ms measurement time per amino acid, pH 2.5, shielding factor $\exp\left(-0.2\sqrt{d}\right)$ (see Methods), initial SNR 10, and bound side chains. Fig 8 shows peptide recovery rates per peptide length and correctly identified peptide counts with these parameters. All other simulations in this section vary only a single parameter from this baseline. The baseline scenario shows a very good peptidome recovery rate (90.69%). With increased measurement times of 1000 ms per amino acid, also most short peptides can be correctly identified and the overall peptidome recovery rate increases further (96.69%). In contrast, shorter measurement times (10 ms) lead to less accurate identification of peptides of all sizes (peptidome recovery rate 73.82%). Fig 8 also shows the negative impact of the presence of side chain signals on the ability to correctly identify peptides with the current identification method. In principle, the superposed side chain signal could be treated as additional information that can be leveraged to improve identification accuracy. We did not pursue this idea further and we treat the superposed signal originating from free side chains as noise.

Fig 9 shows peptide identification results per peptide length with shielding parameter values of 0.1, 0.2, and 0.4. In principle, the shielding parameter is a property of the system and can only be controlled to a limited extent. The baseline shielding parameter $\chi = 0.2$ is an approximation derived through the formula $\sqrt{s \cdot p \cdot 2}/\lambda_D$, in which $s$ is the amino acid size, $p$ is the peptide persistence length, and $\lambda_D$ is the Debye length in room temperature water with a salinity of 10mM. Halving the shielding parameter $\chi$ from 0.2 to 0.1 does not have a significant effect on the identification accuracy, except for a small improvement for the longest peptides (>30 amino acids). However, when increasing $\chi$ to 0.4, the shielding becomes very strong and the peptidome recovery rate is greatly reduced (23.61%).

Peptide identification results per peptide length with initial signal-to-noise ratios of 100, 10, 5, and 2 are shown in Fig 10. With an SNR of 100, peptide identification is slightly better than in the baseline scenario with an SNR of 10. Lowering the SNR has a bigger impact: an SNR of 5 significantly lowers the accuracy of the peptide identification, and with an SNR of 2, almost none of the peptides can be correctly identified. As such, an SNR of at the very least 5 is a solid target for the reprotonation-deprotonation device development.

This target SNR (5) is further explored in Fig 11, by extending the measurement time to 1000 ms and 10 s. As seen in Fig 8, a longer measurement time per amino acid can improve the peptidome recovery rate significantly. With SNR 5, the peptidome recovery rate improves from 45.66% at 100 ms to 70.97% at 1000 ms and to 87.69% at 10 s measurement time per amino acid. Measuring for even longer could potentially further improve peptide identification, but this was not simulated due to excessive computation costs.

Table 2 shows an overview of the proteome recovery rate of all simulations. Since correctly identified peptides lead to correctly identified proteins, there is a strong correlation between the peptidome recovery rates (PeRR) and proteome recovery rates (PRR). A significant fraction of the proteome can however not be identified, because their terminal peptides are not unique in the proteome. As such, Table 2 also shows the "unique PRR", i.e. the PRR limited to those proteins that can be uniquely identified based on their N-terminal tryptic peptides. This metric is closely related to the peptidome recovery rate, but is limited to the peptides that are relevant for protein identification. The best possible protein recovery rate

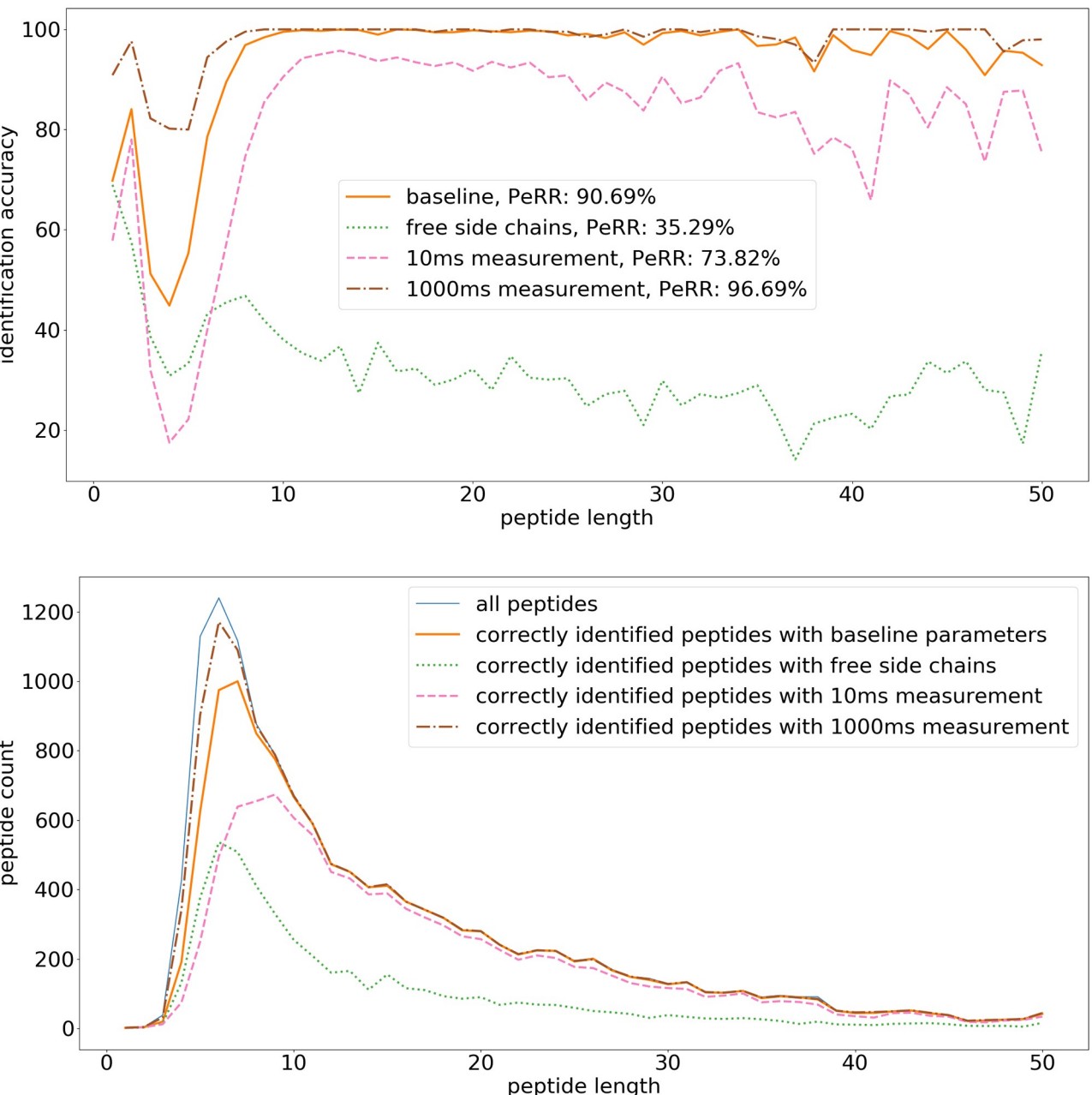

**Fig 8. Peptide identification fraction: Baseline scenario, free side chains, and shorter and longer measurement time per amino acid.** (top) The fraction of correctly identified peptides for each peptide length. The legend additionally contains the peptidome recovery rate (PeRR). (bottom) The total peptide count for each peptide length, and the count of correctly identified peptides. The simulation uses the following parameters: 100 ms measurement time per amino acid, pH 2.5, shielding factor $\exp(-0.2\sqrt{d})$, initial SNR 10, and bound side chains, i.e. additional molecules are attached to the side chains, and no side chain signal is generated. In these simulations three variants on this baseline scenario are also included: (1) free side chains is the scenario of unbound side chains that can freely emit a signal, (2) the measurement time per amino acid is reduced to 10 ms instead of 100 ms, and (3) the measurement time per amino acid is increased to 1000 ms instead of 100 ms.

when identifying proteins based on their N-terminal tryptic peptides corresponds with a unique PRR of 100% and a PRR of 65.42% for this data set. The results show that blocking side chains, longer measurement times per amino acid, weaker shielding and a higher SNR all result in a higher peptide and proteome recovery rates. In particular, blocking side chains

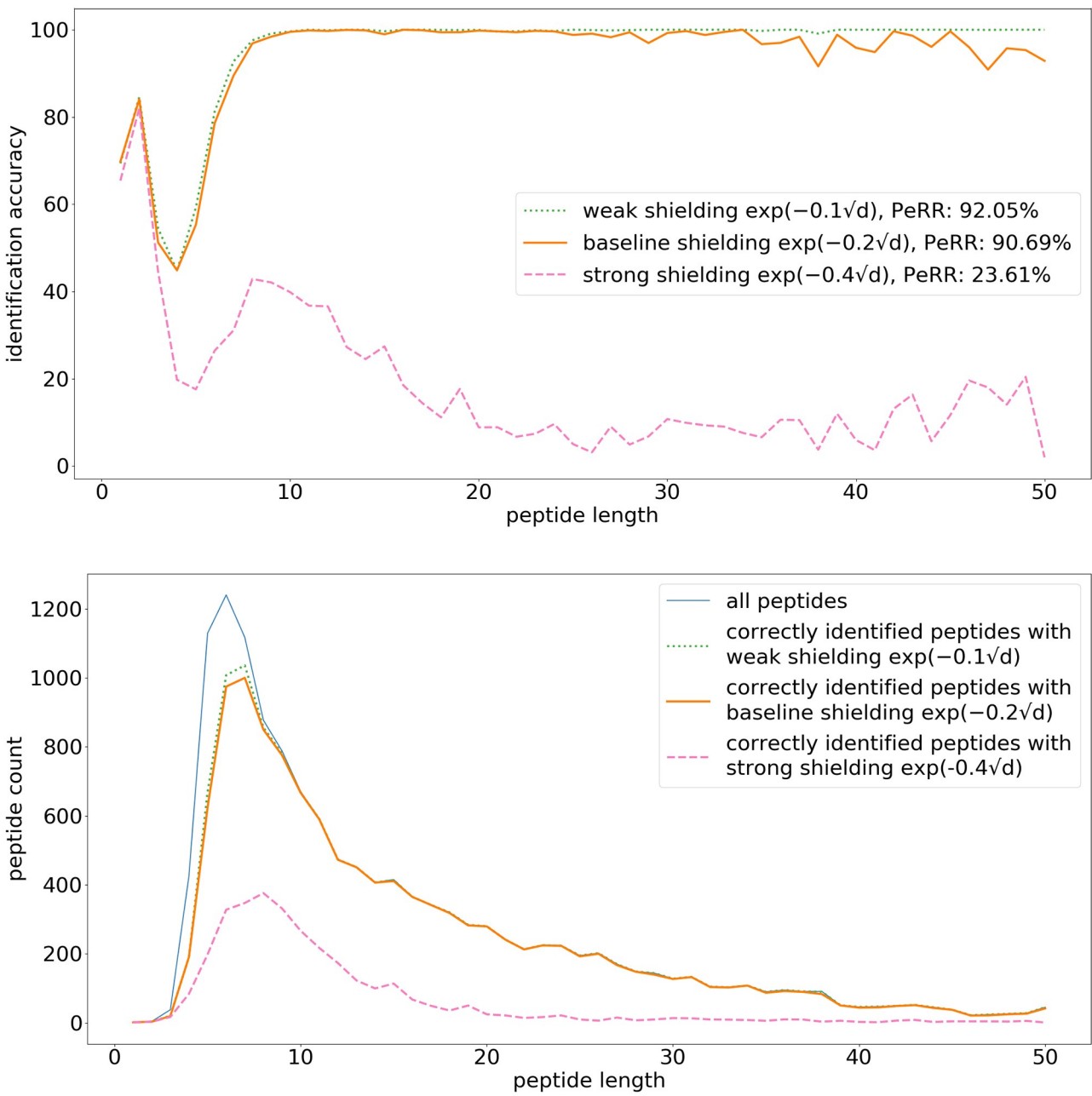

**Fig 9. Peptide identification fraction: Shielding factor.** (top) The fraction of correctly identified peptides for each peptide length. The legend additionally contains the peptidome recovery rate (PeRR). (bottom) The total peptide count for each peptide length, and the count of correctly identified peptides. The simulation uses the following parameters: 100 ms measurement time per amino acid, pH 2.5, initial SNR 10, and bound side chains. In this simulation the shielding factors are respectively $\exp(-0.1\sqrt{d})$, $\exp(-0.2\sqrt{d})$, and $\exp(-0.4\sqrt{d})$, the other parameters are the same as in the baseline Fig 8.

dramatically increases the unique PRR from 35.37% to 93.35%. This is equivalent to 10× fewer unidentified proteins among those that are uniquely identifiable. Similarly, increasing the measurement time per amino acid from 10 ms to 100 ms increases the unique PRR from 76.64% to 93.35%, and decreasing the shielding factor from $\exp(-0.4\sqrt{d})$ to $\exp(-0.2\sqrt{d})$ increases the PRR from 15.26% to 93.35%. An even larger increase in unique PRR (1.92% to

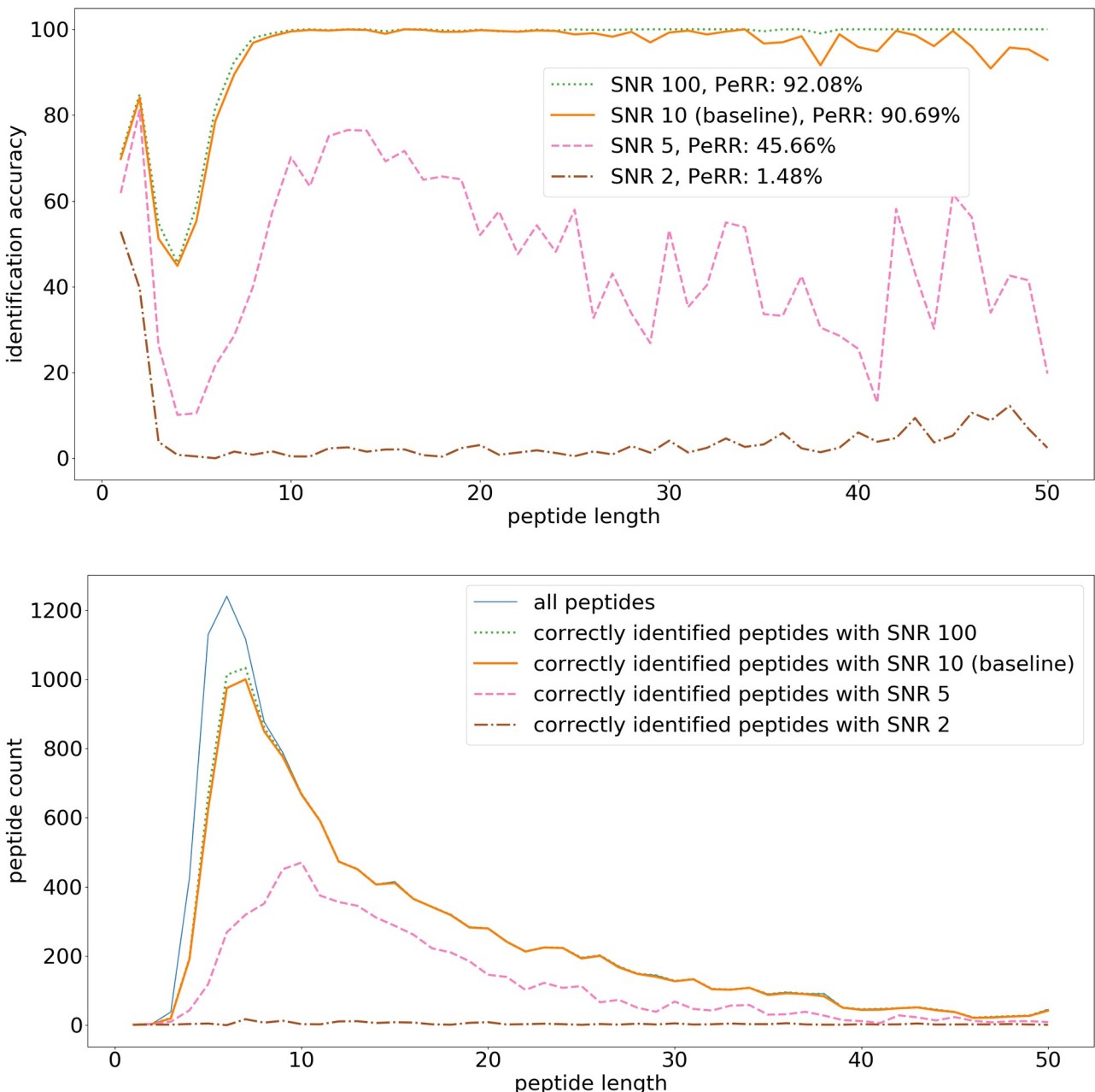

**Fig 10. Peptide identification fraction: Initial SNR.** (top) The fraction of correctly identified peptides for each peptide length. The legend additionally contains the peptidome recovery rate (PeRR). (bottom) The total peptide count for each peptide length, and the count of correctly identified peptides. The simulation uses the following parameters: 100 ms measurement time per amino acid, pH 2.5, shielding factor $\exp(-0.2\sqrt{d})$, and bound side chains. In these simulations the initial SNR are respectively 100, 10, 5, and 2, the other parameters are the same as in the baseline Fig 8.

93.35%) can be seen when increasing the initial SNR from 2 to 10. Further increasing the initial SNR to 100 only provides a minor improvement in PRR. The positive effect of longer measurement times on identification accuracy is further shown in the settings with initial SNR 5 and measurement times 100 ms, 1000 ms, and 10 s, raising the unique PRR from 48.25% to 74.23% and 87.09%, respectively.

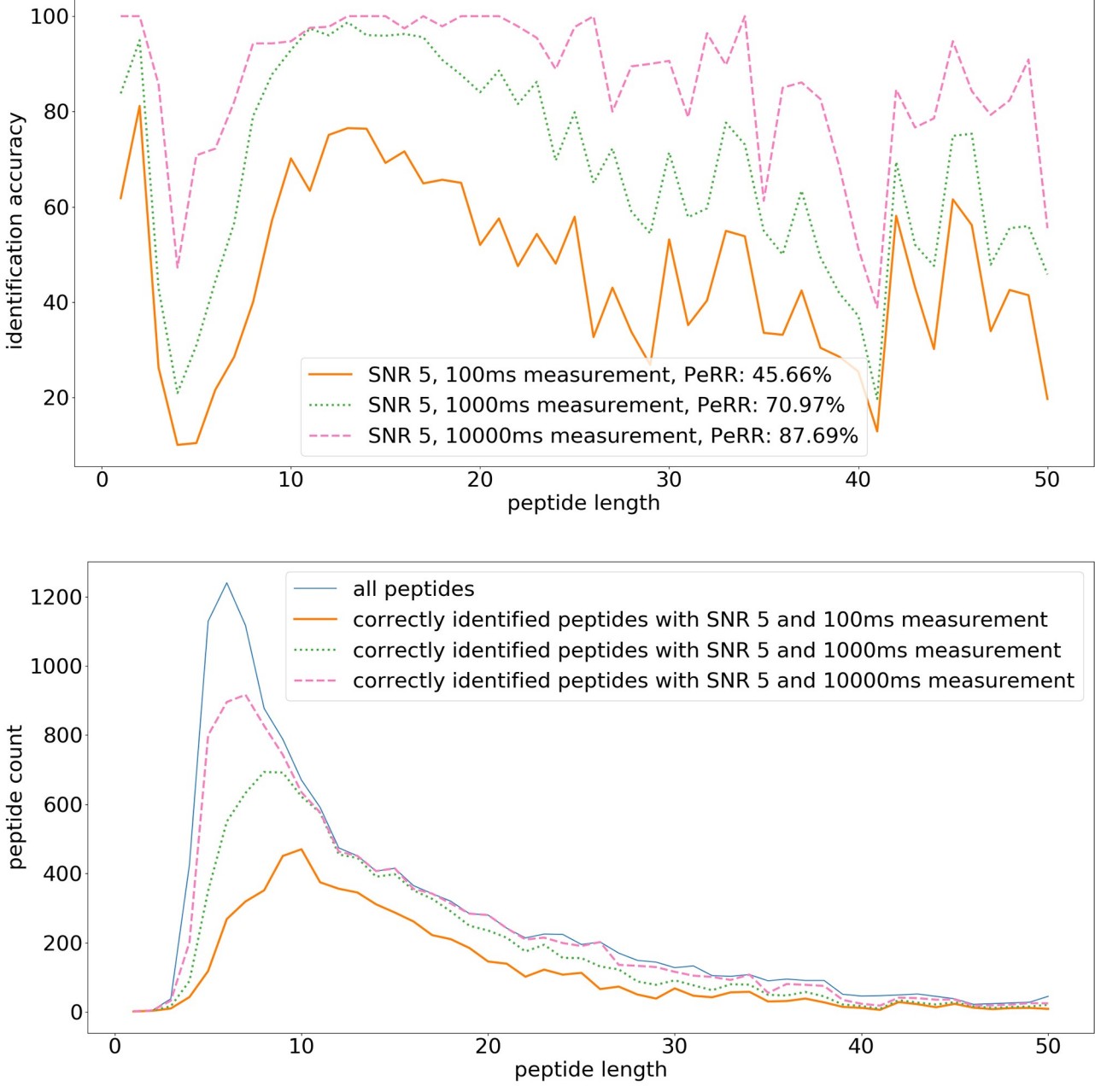

**Fig 11. Peptide identification fraction: Initial SNR 5 with long measurement time.** (top) The fraction of correctly identified peptides for each peptide length. The legend additionally contains the peptidome recovery rate (PeRR). (bottom) The total peptide count for each peptide length, and the count of correctly identified peptides. The simulation uses the following parameters: pH 2.5, shielding factor $\exp(-0.2\sqrt{d})$, and bound side chains. In these simulations the initial SNR is 5 and the measurement times are respectively 100 ms, 1000 ms, and 10 s per amino acid, the other parameters are the same as in the baseline Fig 8.

Empty entries in the parameter columns indicate that the same values were used as in the baseline simulation. "PRR" is the proteome recovery rate over the entire proteome, "Unique PRR" is the proteome recovery rate limited to those proteins that have a unique N-terminal tryptic peptide. Proteins that do not have a unique N-terminal tryptic peptide can never be correctly identified using this method.

**Table 2. Overview of Proteome Recovery Rates (PRR) for different simulation parameters.**

|  | Side chains | Time per amino acid | Shielding factor | Initial SNR | PRR | Unique PRR |
|---|---|---|---|---|---|---|
| Baseline | blocked | 100 ms | $\exp(-0.2\sqrt{d})$ | 10 | 61.07% | 93.35% |
| Side chains | free |  |  |  | 23.14% | 35.37% |
| Time per amino acid |  | 10 ms |  |  | 50.51% | 77.21% |
|  |  | 1000 ms |  |  | 64.49% | 98.59% |
| Shielding factor |  |  | $\exp(-0.1\sqrt{d})$ |  | 61.88% | 94.59% |
|  |  |  | $\exp(-0.4\sqrt{d})$ |  | 11.30% | 17.27% |
| Initial SNR |  |  |  | 100 | 61.71% | 94.33% |
|  |  |  |  | 5 | 31.57% | 48.25% |
|  |  |  |  | 2 | 1.25% | 1.92% |
| SNR 5 and time |  | 1000 ms |  | 5 | 48.57% | 74.23% |
|  |  | 10 s |  | 5 | 56.98% | 87.09% |

## Conclusion

We propose protonation-based protein sequencing, which identifies the terminal amino acid by measuring the alternation between protonated and deprotonated states of the terminal amino acid. Unlike other recently proposed methods, this method does not rely on optical imaging. As a result, this method does not suffer from loss of accuracy due to inefficient labeling or photobleaching. However, other disturbances of the signal occur. First, side chain reprotonation-deprotonation emits a secondary signal with a higher amplitude than the main signal. This leads to a significant decrease in accuracy with the identification method that we applied in this work. More sophisticated techniques could be developed to accurately discern the main signal from the side chain signals. As a matter of fact, the side chains could actually contribute to a more precise identification of the amino acids in the peptide, at a higher data processing cost. Second, the shielding of the electrical signal by the electrolyte can have a significant impact on the identification accuracy of long peptides. The first measurements associated with amino acids at a long distance from the peptide anchoring point result in a very weak signal. Still, the later measurements in these peptides occur near to the binding point and thus yield stronger signals. The identification method could be further improved by taking this into account and assigning a higher weight to the most accurate signals. Furthermore, the signal-to-noise ratio has a major impact on the ability to correctly discern peptides, as evidenced by a proteome recovery rate (PRR) of 61.71% with SNR 10 and a PRR of 31.57% with SNR 5, both with 100 ms measurement time per amino acid. Longer measurement times per amino acid lead to accurate estimations of the on-off-signal. Therefore, even with a relatively low initial SNR of 5, the proteome recovery rate can be increased significantly by increasing the measurement time per amino acid: from PRR 31.57% at 100 ms to PRR 56.98% at 10 s.

In this work, we implemented a simulator that takes these disturbances into account, however, some additional biases and sources of noise may be present in an actual realization of a reprotonation-deprotonation protein sequencing device. We have shown that a noisy reprotonation-deprotonation signal can result in highly accurate protein identification, by using many repeated measurements to get an accurate estimation of the terminal amino acid pKa value. A successful implementation of this sequencing method, with a sufficiently high signal-to-noise ratio, e.g. 5, will require further advances in chip design. However, the realization of this sensor can lead to satisfactory proteome identification as put forward in this work. Furthermore, the fast and accurate determination of protein sequences with reprotonation-deprotonation sequencing can potentially scale to massively parallel sequencing with a high

throughput. As such, the successful execution of this sequencing technique will elevate proteomics to the level of reference-based genomics, with high throughput and sensitivity. With further improvements to the basic peptide identification scheme shown here, the reprotonation-deprotonation protein sequencing method could also be used for accurate de novo desequencing of peptides and proteins.

## Methods

In this section, the simulation approach is described. Proteins from a database are digested *in silico* into peptides, and for these peptides a signal set is generated. This electrical signal could be stored in its entirety and used for identification. However, at a rate of 10 MHz and a 4 byte floating point value per data point, this results in 40 MB of data per second for each signal. This could be prohibitive when scaling this method to large samples with many proteins. Another approach is to only store summary statistics of the signal. Here, the signal is processed in chunks of a fixed duration (e.g. 1 ms), for which the average on-fraction is computed. Only this on-fraction is stored, resulting in a sizable reduction in required bandwidth and storage. To compute the on-fraction, first the window is processed with a median filter (default filter size of 3) to remove any obvious outliers. Then, the average of the maximal and minimal voltage in the filtered window is used as a threshold value, and the fraction of data points with a higher voltage is the on-fraction. This results in a list of on-fractions per amino acid, and due to the short duration of each chunk, this list may cover a wide range of values. The final on-fraction for the amino acid is then obtained by taking the median value from this list of on-fractions.

To identify peptides, the signal is first processed as described above: the median on-fraction of chunks is kept for each amino acid. For each amino acid the expected on-fraction is calculated based on the pKa and pH, and this information is then used to determine the most likely peptide in a reference database that resulted in the given signal. Proteins can be identified based on peptides unique to that protein, i.e., if a peptide only occurs in a single protein of the reference database. If such a peptide is identified, then the corresponding protein, or a novel protein, must be present in the original sample.

In the presence of side chains, the measured on-fraction can be affected. If the pH of the solution is significantly higher or lower than the pKa of the side chain, then the side chain is almost always in a fixed state, i.e. protonated or deprotonated, and this is not an issue. The only way to lock all side chains in this manner without also removing the main reprotonation-deprotonation signal, is by selecting a very acidic pH ($<1$) and measuring the reprotonation-deprotonation of the carboxyl group. Such a low pH is unlikely to be feasible in practice. However, since there are basic and acidic side chains, at any given pH, at least some of them are unlikely to change states in any given measurement at a fixed pH. An alternative approach is to effectively block the sidechain reprotonation-deprotonation signal by binding another molecule to it. In practice, these blocking molecules do not have 100% binding efficiency, and some side chain reprotonation-deprotonation signals will remain present in the measured signal.

### Data

Simulations were performed using H. sapiens protein data retrieved through UniProt. The reviewed Swiss-Prot data was used, rather than the unreviewed TrEMBL data. In total, this amounts to 19,886 H. sapiens proteins. Protein sequences were converted *in silico* to peptides with an endopeptidase, e.g. trypsin, cyanogen bromide, or Glu-C.

## Evaluation metrics

Monte Carlo simulations of sequencing experiments provide artificial measurement data from which the original peptide is then identified. Several metrics are used to evaluate the identification accuracy. All reported metrics are averaged over 1000 simulation runs. The peptidome recovery rate (PeRR) is the ratio of correctly identified peptides and the total number of generated signals, for the entire peptide database. The proteome recovery rate (PRR) is determined in an analogous manner as the ratio of the correctly recovered proteins and the total number of simulated proteins. Note that only N-terminal and C-terminal peptides are used to identify proteins. As some proteins do not have a unique terminal peptide, they cannot be uniquely identified. Such cases are counted as incorrectly identified proteins.

## Simulation process

On-times and off-times are simulated based on the pH value of the solution and the pKa values of the amino acids. The pKa values for the amine groups, carboxyl groups and side chains that are used in the simulations are shown in Table 1. Different pKa values have been reported for the same amino acids [30]. In practice, the pKa of amino acids are site-specific, i.e. they depend on the location of the peptide in the protein [31]. Using a more advanced identification algorithm this can be taken into account by learning the pKa values based on known peptide databases.

The electrical signal is simulated, taking into account several sources of noise. The individual components whose sum constitutes the simulated signal are:

1. the reprotonation-deprotonation of the terminal amino acid,

2. the reprotonation-deprotonation of any side chains in the peptide, and

3. the noise inherent to the system.

The relative power of these different subsignals is determined by two simulation parameters: (1) the initial signal-to-noise ratio (SNR) $S$, and (2) the shielding factor $\exp(-\chi f(d))$. The initial SNR $S$ is the SNR for a reprotonation-deprotonation event at a distance 0 of the anchor point of the peptide. The signal amplitude decreases with the distance from the anchor point as controlled by the shielding factor. For the $d$th amino acid, starting from the anchor point, the average amplitude of the reprotonation-deprotonation signal is given by $S\exp(-\chi f(d))$, with $f$ some monotonically increasing function, and $\chi$ a constant positive shielding parameter. This effectively reduces the SNR for amino acids that are further removed from the anchor point.

**Underlying terminal reprotonation-deprotonation signal.** The underlying reprotonation-deprotonation signal consists of consecutive low and high voltages, where the change in voltage is approximately proportional to the change in charge at the protonation site on the peptide. The low voltage corresponds with an off-state (-COO$^-$ or -NH$_2$) and the high voltage corresponds with an on-state (-COOH or -$^+$NH$_3$). Note that the voltage difference between the on-state and the off-state is considered. In particular, the high voltage state for the carboxyl group is the neutral state (-COOH), while the low voltage state is the negatively charged state (-COO$^-$). The transition rates between both states are called $k_{on}$ and $k_{off}$. Here, $k_{on}$ is the transition rate from off-state to on-state in $Hz$, and as such the expected off-time is $1/k_{on}$ seconds. Analogously $k_{off}$ is the transition rate to the off-state, and the expected on-time is $1/k_{off}$ seconds. By definition of the pKa, if pH = pKa, then $k_{off} = k_{on}$ and the system will be in the on-state for 50% of the time. In these simulations, the average off-time is fixed at 1MHz [32]; depending on the sequencing environment other values might be more appropriate. If this

value is changed, and the signal rate and sequencing time are scaled accordingly, then all simulation results are unaffected. Together with the pKa and pH of the solution, this leads to the following transition speeds between states:

- on → off: $k_{off} = 10^{6+\ \mathrm{pH-pKa}}$ Hz,

- off → on: $k_{on} = 10^6$ Hz.

If the pH is decreased (resp. increased), i.e. the solution becomes more acidic (resp. basic), then the system will be in the on-state for a larger (resp. smaller) fraction of the time. This is reflected in the formula for $k_{off}$: a lower pH decreases the transition speed from on to off. On- and off-times are the amount of time between events in a Poisson point process, i.e. the association and dissociation events occur continuously, independently and at a constant average rate ($k_{on}$ and $k_{off}$). As a result these times are exponentially distributed as follows:

1. on-time $\sim$ Exp($k_{off}$), and

2. off-time $\sim$ Exp($k_{on}$),

where Exp($k$) is the exponential distribution with rate parameter $k$ and probability density function $f(x) = ke^{-kx}$. The on-time distribution has rate parameter $k = k_{off}$, while the off-time distribution has rate parameter $k = k_{on}$. The underlying terminal reprotonation-deprotonation signal is then simulated by alternately drawing from the on- and off-distributions. Fig 12 shows an example signal obtained in this manner.

The amplitude of the signal in Fig 12 is primarily based on the shielding factor $\exp(-\chi f(d))$ and the distance of the amino acid to the anchor point. On top of this shielding, there is a source of normal noise added to the amplitude of each state. This extra uncertainty is related to for example the movement of the peptide in the solution. The amplitude as a function of (1) $d$ the 1-based index of the terminal amino acid, (2) the shielding factor $\exp(-\chi f(d))$, and (3) the normal noise standard deviation $\sigma$, is given by:

$$A = sN(\exp(-\chi f(d)), \sigma^2).$$

In the simulations $\sigma$ is always fixed to 0.1.

**Sidechain reprotonation-deprotonation signal.** The sidechain reprotonation-deprotonation signals are entirely analogous to the terminal reprotonation-deprotonation signal, the only differences are (1) the on-times are determined by the pKa of the sidechain, and (2) the average amplitude depends on the location of the sidechain in the peptide, i.e. it does not necessarily originate from the terminal amino acid position. Sidechains can have acidic or basic pKa values.

## Signal rate

Another simulation parameter is the signal rate $r$, by default $r = 10\mathrm{MHz}$. This rate determines the time between each sample of the signal. If the sampling rate is too low events can be missed. If for example a certain reprotonation-deprotonation signal switches on and off between two sample points, then the resulting on-state will not be present in the signal. The probability of missing an event depends on (1) the duration of the event, and (2) the sampling rate $r$. Depending on the method that is used to identify the signal, missing events do not have to be inherently problematic. When identification is based on average on-time, and on- and off-states are accurately identified, then sampling for a longer time will always result in a better approximation of the true average on-time. However, for a lower sampling rate $r$ a longer sampling time will be required to achieve the same accuracy.

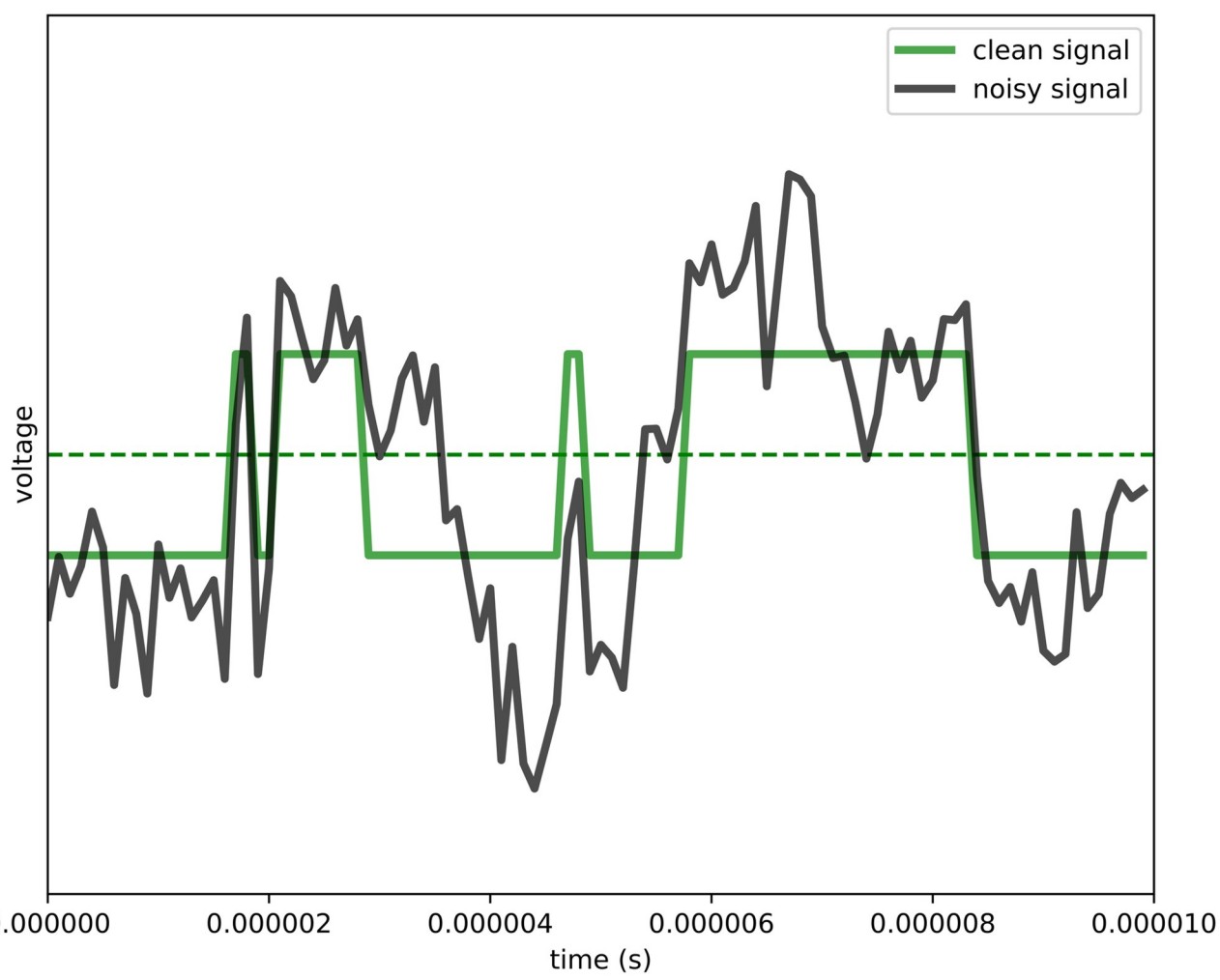

**Fig 12. Example of a simulated on-off-signal with the addition of 1/*f* noise (black) and the underlying on-off-signal (green).**

### 1/*f* Noise

The electrical noise is assumed to be 1/*f* noise, also known as *flicker noise* and *pink noise*. The noise is simulated with the Voss-McCartney algorithm. This algorithm adds several sources of white noise to emulate pink noise. The pink noise is then rescaled linearly to have the desired power, as prescribed by the SNR parameter *S*. Fig 12 shows an example signal with the additional inclusion of 1/*f* noise. The figure shows how noise can lead to false on- and off-events. Further sources of noise can occur in practice, thermal noise for example has not been included in these simulations.

## Author Contributions

**Conceptualization:** Giles Miclotte, Koen Martens.

**Methodology:** Giles Miclotte, Koen Martens, Jan Fostier.

**Software:** Giles Miclotte.

**Supervision:** Koen Martens, Jan Fostier.

**Validation:** Giles Miclotte.

**Visualization:** Giles Miclotte.

**Writing – original draft:** Giles Miclotte.

**Writing – review & editing:** Giles Miclotte, Koen Martens, Jan Fostier.

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
