## [Decision Letter · Decision Letter 0]

13 Jan 2020

PONE-D-19-32444

Computational assessment of the feasibility of protonation-based protein sequencing

PLOS ONE

Dear Dr. Miclotte,

Thank you for submitting your manuscript to PLOS ONE. After careful consideration, we feel that it has merit but does not fully meet PLOS ONE’s publication criteria as it currently stands. Therefore, we invite you to submit a revised version of the manuscript that addresses the points raised during the review process.

The main concerns from the reviewers are the feasibility of the proposed method and the protein data base used in this paper. We can reconsider the manuscript if the authors can satisfactorily address all the concerns raised by the reviewers and improve the writing of the manuscript.

We would appreciate receiving your revised manuscript by Feb 27 2020 11:59PM. To enhance the reproducibility of your results, we recommend that if applicable you deposit your laboratory protocols in protocols.io, where a protocol can be assigned its own identifier (DOI) such that it can be cited independently in the future. For instructions see: http://journals.plos.org/plosone/s/submission-guidelines#loc-laboratory-protocols

We look forward to receiving your revised manuscript.

Kind regards,

Wenfei Li, Ph.D.

Academic Editor

PLOS ONE

Journal Requirements:

2. Thank you for stating the following in the Financial Disclosure section:"The authors received no specific funding for this work."

Thank you for stating the following in the Competing Interests section:"The authors have declared that no competing interests exist."

We note that one or more of the authors are employed by a commercial company: 'imec, Leuven'

b). Please also provide an updated Competing Interests Statement declaring this commercial affiliation along with any other relevant declarations relating to employment, consultancy, patents, products in development, or marketed products, etc. 

Within your Competing Interests Statement, please confirm that this commercial affiliation does not alter your adherence to all PLOS ONE policies on sharing data and materials by including the following statement: "This does not alter our adherence to  PLOS ONE policies on sharing data and materials.” (as detailed online in our guide for authors http://journals.plos.org/plosone/s/competing-interests) . If this adherence statement is not accurate and  there are restrictions on sharing of data and/or materials, please state these. Please note that we cannot proceed with consideration of your article until this information has been declared

Reviewers' comments:

Reviewer's Responses to Questions

**Comments to the Author**

1. Is the manuscript technically sound, and do the data support the conclusions?

Reviewer #1: Partly

Reviewer #2: Partly

Reviewer #3: Partly

2. Has the statistical analysis been performed appropriately and rigorously? 

Reviewer #1: No

Reviewer #2: N/A

Reviewer #3: Yes

3. Have the authors made all data underlying the findings in their manuscript fully available?

Reviewer #1: Yes

Reviewer #2: Yes

Reviewer #3: Yes

4. Is the manuscript presented in an intelligible fashion and written in standard English?

Reviewer #1: No

Reviewer #2: Yes

Reviewer #3: Yes

5. Review Comments to the Author

Reviewer #1: The authors used only known protein data base which is very tricky and dangerous. At least the simulator should show capability to work with heterogeneous samples or unknown pure sample.

The authors used endopeptidase to convert the proteins into peptides. However, throughout the main content, there is no information or reference how this enzyme work with the peptide. The authors should provide much more detail.

In Result and discussion section, the authors stated “A mixture of proteins is digested into peptides using an endopeptidase.” What is the contained in this mixture, what is the ratio and concentration? Throughout the whole manuscript, there is a lack of description of the working condition, how is the reliability when comparing to the experimental work?

pH condition is one of the main approaches to determine the protonation time spent. As most widely known, under different pH condition, the protein or peptide refers to totally different biological behavior and even inducing dysfunctionalization.

The authors focused on the carboxyl group to eliminate the influence from the side chain. There are very close pKa (Threonine: 2.09 and Aspartic Acid: 2.10, Glutamic Acid: 2.10). How the authors differentiate each of them via the protonation time readout?

The authors adjusted the sojourn of each amino acid readout, at 10 ms/base the accuracy is 32.94% when the read speed lowers to 1000 ms/base, the accuracy increased to 79.50% there are plenty of room to increase the accuracy. One approach addressed by the authors is to use an adequate SNR, is there any other comments by unlimitably lower the reading rate or play around the other physical parameters?

Reviewer #2: Comments to the Authors for the manuscript: PONE-D-19-32444

1. In Page 3/16, in the section ‘Protonation-based protein sequencing’, the N-terminal amine is wrongly depicted as (-CH2). The N-terminal amine must be (-NH2). And therefore, in the reaction shown in the following line: the N-terminal amine: -CH3+ must be corrected to –NH3+ and –CH2 should be corrected to –NH2.

2. In pages 3/16 (last line) and 4/16 (first two lines): the authors mention that they identify amino acids through their carboxyl group pKa values. This means that upon ‘Deprotonation’, the carboxyl group would possess Negative charge and on ‘Reprotonation’, the carboxyl group would NOT be charged, viz., Neutral. Further, the authors have defined that ‘Deprotonation’ leads to “OFF” position and ‘Reprotonation’ leads to “ON” position (Figure 3). So, does this mean that formation of ‘NEGATIVELY CHARGED Carboxyl group’ due to Deprotonation, corresponds to OFF state, while formation of ‘Neutral or Uncharged carboxyl group’ due to Reprotonation, corresponds to ON state? Can authors clarify this?

3. In Page 5/16, in the ‘Results and Discussion’ section, the authors define the ON and OFF states: “The signal consists of a high voltage, when a proton is bound (ON-state) and a low voltage otherwise (OFF-state) (see Figure 3).” Since, the authors have opted N-terminus of the peptide for anchoring to the substrate, the Carboxyl group of the peptide is Free. And in the case of Free Carboxyl group, the Proton Bound state (i.e., Reprotonation) is actually UNCHARGED, viz., -COOH. So, it is not clear, as to, how the authors define or consider the UNCHARGED Proton Bound form: -COOH, to be ON-state. In other words, how the UNCHARGED form can contribute for high voltage signal? Likewise, it is not clear, how the Deprotonated form of carboxyl group, -COO-, which has a single Negative charge can contribute for low voltage, i.e., OFF-state. In other words, it is not understandable, as to, how the authors correlate the CHARGED Form (-COO-) to OFF-state (low voltage) and UNCHARGED Form (-COOH) to ON-state (high voltage).

Moreover, in lower pH range of 2 – 5, the population of UNCHARGED form (-COOH) of Carboxyl group (-COOH) would be more than the CHARGED form (-COO-), i.e., the UNCHARGED Form would have a greater lifetime than the CHARGED form in lower pH range of 2 - 5. So, would it mean that in lower pH range of 2 – 5, higher frequency of ON-states or higher voltages would be detected due to UNCHARGED form (-COOH), as per the authors’ current definition of ON and OFF-states? If this is the case, how is it possible for the UNCHARGED form (-COOH) to contribute for higher voltage/ON-state? In contrast, when the population or the lifetime of UNCHARGED form (-COOH) is higher, then it should contribute for lower voltage or OFF state, because of absence of charge on the molecule.

Can the authors explain or rationalize this clearly? Without proper explanation or understanding of ON and OFF states, this study would not have any meaning or further applications. So, Figure 3 may require modification, in that, the definitions of ON/OFF states for Free Carboxyl group can be different from the ON/OFF states for Free Amino group, depending on the pH of the medium, in which the peptide is present.

4. In Page 13/16, in the section “Underlying terminal reprotonation-deprotonation signal”, from the equations defined for ‘kon’ and ‘koff’, it can be understood that

(i) Larger the difference between pH and pKa would correspond to Lower ‘koff’ values, which implies that the peptide would be in the ON-state for a Larger fraction of time.

(ii) Lower the difference between pH and pKa would mean Higher ‘koff’ values, which implies that the peptide would be in the OFF-state for a Larger fraction of time.

It is well known that the pKa values of free Carboxyl group of C-terminus of the peptide and the pKa values of sidechain carboxyl groups of Aspartic acid (Asp) and Glutamic acid (Glu) are very close to the acidic pH value. And hence, ALL the carboxyl groups would be in the OFF-state for a Larger time fraction. Moreover, at lower pH values, the population of UNCHARGED form (i.e. Deprotonated form) of the carboxyl groups would also be Higher. So, according to the equation proposed by the authors, it is clear that the representation of Protonation and Deprotonation is not correct, as shown in Figure 3. Therefore, Figure 3 must be modified, so as to have agreement with equation proposed for ‘koff’. In other words, the authors have defined ‘koff’ and ‘kon’equations correctly, but these equations do not accurately represent the Proton-bound form and the Deprotonated form, particularly for the free Carboxyl groups.

Overall, the authors are urged to provide clarification for the proper definition of ON-state and OFF-state, by taking into account the conditions that are shown below:

(i) for Free Carboxyl Groups (whose pKa values are closer to acidic pH):

Deprotonation = NEGATIVELY CHARGED Form = -COO for pH > ~ 2.5 or 3 and

Reprotonation = UNCHARGED Form = -COOH for pH < ~ 2

Therefore, Deprotonation = ON-state (Higher Voltage) and

Reprotonation = OFF-state (Lower Voltage)

(ii) for Free Amino Groups (whose pKa values are higher than acidic pH):

Deprotonation = UNCHARGED Form = -NH2 for pH > 10 (approx.) and

Reprotonation = POSITIVELY CHARGED Form = -N+H3 in the pH range: 2 – 10.

Therefore, Deprotonation = OFF-state (Lower Voltage) and

Reprotonation = ON-state (Higher Voltage)

Thus, the representation shown in Figure 3 is correct for Free Amino Groups and not for Free Carboxyl Groups. The authors need to make a note of this and do the necessary corrections.

5. Digestion of proteins would give rise to many peptides. So, how is it ensured that only One Molecule of the Peptide binds to a chip? Is the chip designed in a manner that only One Molecule of the Peptide can be accommodated on its surface? It is established that chromatographic methods aid in decreasing the complexity of the mixture of peptides. So, before allowing the digested peptides to bind onto the chip surface, is the mixture of digested peptides subjected to any chromatographic method? In other words, are the peptides introduced to bind onto the chip surface, after any chromatographic method? Or, is the mixture of digested peptides directly introduced onto the chip surface for binding?

6. The authors need to mention the source, from where the pKa values have been taken to prepare Table 1.

7. What do the authors mean by ‘clusters’? It is better to give a proper definition for ‘clusters’ and then describe about them. How a cluster is different from sequence?

8. Why the authors have not compared their simulation results with some experimental data that may be already available?

Reviewer #3: In this manuscript, the authors propose a new single-molecule protein sequencing method. The scheme is based on measurements of the average time an amino acids spends in a protonation states using field effector transistor (FET). The concept is novel, which makes the manuscript suitable for publication in Plos One. But the feasibility of this method is not high, so the authors should answer several concerns for the publication.

1) The authors claim that a carbon nanotube FET can recognize the gate electric field generated by a single charge. But it is unlikely that the gate electric field generated by a single charge can give such a significant effect on the carbon nanotube channel in a real experiment. There are many other factors contributing to fluctuations in the electrical signal, such as electrical noise, thermal fluctuations, and electrolyte ion migration, which would result in larger effects. This will be much more extreme than the cases in the references (17-22) since the proposed device measures only a single charge. This concern should be addressed in the manuscript.

2) Related to (1), a 2004 paper (ref 16) is cited where a single peptide was measured. Yet in this reference, a very simplified model peptide was used, where only one amino acid side chain group could dissociate. It is not obvious how signals could be resolved when multiple dissociating groups would present when analysing peptides derived from proteins. This concern should be addressed in the manuscript.

3) Authors discuss the read error from shielding, which stems from the significant distance from the peptide end to the surface (effect related to the measuring device). However, one can expect that local intramolecular interaction (hydrogen bonding etc.) will also affect the pKa of particular amino acid residues. This error of pKa variance is not modelled or discussed.

4) It would be challenging to make a sensor that is able to withstand repeatedly chemicals used in Edmann degradation or one where proteins would not get adsorbed non-specifically. Please discuss these issues.

Minor comments

1) Figure 1 does not accurately describe how detection would be achieved. Please revise it.

2) In page 3, the deprotonation scheme is incorrect for N-terminal amine: -CH3+->CH2 + H+. Carbon should be replaced with nitrogen.

3) pKa values of dissociating groups of amino acids are different among literature. The authors should comment on their choice.

6. PLOS authors have the option to publish the peer review history of their article (what does this mean?). If published, this will include your full peer review and any attached files.

Reviewer #1: No

Reviewer #2: Yes: Varatharajan Sabareesh

Reviewer #3: Yes: Chirlmin Joo

---

## [Author Response · Author response to Decision Letter 0]

16 Apr 2020

Reviewer #1: 

1) The authors used only known protein data base which is very tricky and dangerous. At least the simulator should show capability to work with heterogeneous samples or unknown pure sample.

The authors used endopeptidase to convert the proteins into peptides. However, throughout the main content, there is no information or reference how this enzyme work with the peptide. The authors should provide much more detail.

In Result and discussion section, the authors stated “A mixture of proteins is digested into peptides using an endopeptidase.” What is the contained in this mixture, what is the ratio and concentration? Throughout the whole manuscript, there is a lack of description of the working condition, how is the reliability when comparing to the experimental work?

Any and all references to the use of peptidase or any other molecules in the manuscript are theoretical, the work described here is entirely in-silico. The simulator is intended as a method to assess the viability of this sequencing method. In particular all peptide sequences are independently processed and individually compared with a reference data set. Unknown samples would require de novo identification, which is discussed in the subsection “Sequencing isolated amino acids” of the results section in the manuscript.

2) pH condition is one of the main approaches to determine the protonation time spent. As most widely known, under different pH condition, the protein or peptide refers to totally different biological behavior and even inducing dysfunctionalization.

We have limited the scope of this method to the scenario where proteins are extracted from their natural environment, denatured, and digested. The focus is entirely on the identification of proteins (or peptides).

3) The authors focused on the carboxyl group to eliminate the influence from the side chain. There are very close pKa (Threonine: 2.09 and Aspartic Acid: 2.10, Glutamic Acid: 2.10). How the authors differentiate each of them via the protonation time readout?

This is indeed one of the main challenges of the proposed sequencing method. De novo it is not possible to differentiate between amino acids with the same pKa values, but it is possible to narrow it down to a small subset of amino acids as shown in the results section (Sequencing isolated amino acids).

The authors adjusted the sojourn of each amino acid readout, at 10 ms/base the accuracy is 32.94% when the read speed lowers to 1000 ms/base, the accuracy increased to 79.50% there are plenty of room to increase the accuracy. One approach addressed by the authors is to use an adequate SNR, is there any other comments by unlimitably lower the reading rate or play around the other physical parameters?

Due to the central limit theorem, a lower read speed will in general result in a higher accuracy in the absence of bias, and this is also supported by the simulation results. A further reduction in read speed was not feasible to simulate due to excessive run times. Additionally, lowering the reading rate does not allow the method to distinguish amino acids with (near) identical pKa values. However, this method can be used to provide accurate ambiguous amino acid calls, e.g. “threonine, or aspartic acid, or glutamic acid”. As shown in subsection “Sequencing Peptides”, this information is sufficient to accurately identify proteins with a protein recovery rate of up to 98.59% at 1000 ms per amino acid at SNR 10, or 87.09% at 10s per amino acid at SNR 5.

Reviewer #2: Comments to the Authors for the manuscript: PONE-D-19-32444

1. In Page 3/16, in the section ‘Protonation-based protein sequencing’, the N-terminal amine is wrongly depicted as (-CH2). The N-terminal amine must be (-NH2). And therefore, in the reaction shown in the following line: the N-terminal amine: -CH3+ must be corrected to –NH3+ and –CH2 should be corrected to –NH2.

This has been corrected in the manuscript.

2. In pages 3/16 (last line) and 4/16 (first two lines): the authors mention that they identify amino acids through their carboxyl group pKa values. This means that upon ‘Deprotonation’, the carboxyl group would possess Negative charge and on ‘Reprotonation’, the carboxyl group would NOT be charged, viz., Neutral. Further, the authors have defined that ‘Deprotonation’ leads to “OFF” position and ‘Reprotonation’ leads to “ON” position (Figure 3). So, does this mean that formation of ‘NEGATIVELY CHARGED Carboxyl group’ due to Deprotonation, corresponds to OFF state, while formation of ‘Neutral or Uncharged carboxyl group’ due to Reprotonation, corresponds to ON state? Can authors clarify this?

The on-state and the off-state are precisely defined by the presence of a bound proton or the lack thereof. As such, deprotonation is the transition from on-state to off-state, and reprotonation is the transition from off-state to on-state. In the case of a carboxyl group this indeed corresponds with a negative charge for the off-state and a neutral charge for the on-state. This has been clarified in the manuscript.

3. In Page 5/16, in the ‘Results and Discussion’ section, the authors define the ON and OFF states: “The signal consists of a high voltage, when a proton is bound (ON-state) and a low voltage otherwise (OFF-state) (see Figure 3).” Since, the authors have opted N-terminus of the peptide for anchoring to the substrate, the Carboxyl group of the peptide is Free. And in the case of Free Carboxyl group, the Proton Bound state (i.e., Reprotonation) is actually UNCHARGED, viz., -COOH. So, it is not clear, as to, how the authors define or consider the UNCHARGED Proton Bound form: -COOH, to be ON-state. In other words, how the UNCHARGED form can contribute for high voltage signal? Likewise, it is not clear, how the Deprotonated form of carboxyl group, -COO-, which has a single Negative charge can contribute for low voltage, i.e., OFF-state. In other words, it is not understandable, as to, how the authors correlate the CHARGED Form (-COO-) to OFF-state (low voltage) and UNCHARGED Form (-COOH) to ON-state (high voltage).

Moreover, in lower pH range of 2 – 5, the population of UNCHARGED form (-COOH) of Carboxyl group (-COOH) would be more than the CHARGED form (-COO-), i.e., the UNCHARGED Form would have a greater lifetime than the CHARGED form in lower pH range of 2 - 5. So, would it mean that in lower pH range of 2 – 5, higher frequency of ON-states or higher voltages would be detected due to UNCHARGED form (-COOH), as per the authors’ current definition of ON and OFF-states? If this is the case, how is it possible for the UNCHARGED form (-COOH) to contribute for higher voltage/ON-state? In contrast, when the population or the lifetime of UNCHARGED form (-COOH) is higher, then it should contribute for lower voltage or OFF state, because of absence of charge on the molecule.

Can the authors explain or rationalize this clearly? Without proper explanation or understanding of ON and OFF states, this study would not have any meaning or further applications. So, Figure 3 may require modification, in that, the definitions of ON/OFF states for Free Carboxyl group can be different from the ON/OFF states for Free Amino group, depending on the pH of the medium, in which the peptide is present.

The neutral voltage of the uncharged on-state (-COOH) is higher than the negative voltage of the charged off-state (-COO-). The pH of the medium has an effect on the transition speeds between the states, but the interpretation of the states themselves is independent of the particular group (carboxyl or amino) and the pH. This has been clarified in the manuscript.

4. In Page 13/16, in the section “Underlying terminal reprotonation-deprotonation signal”, from the equations defined for ‘kon’ and ‘koff’, it can be understood that

(i) Larger the difference between pH and pKa would correspond to Lower ‘koff’ values, which implies that the peptide would be in the ON-state for a Larger fraction of time.

(ii) Lower the difference between pH and pKa would mean Higher ‘koff’ values, which implies that the peptide would be in the OFF-state for a Larger fraction of time.

It is well known that the pKa values of free Carboxyl group of C-terminus of the peptide and the pKa values of sidechain carboxyl groups of Aspartic acid (Asp) and Glutamic acid (Glu) are very close to the acidic pH value. And hence, ALL the carboxyl groups would be in the OFF-state for a Larger time fraction. Moreover, at lower pH values, the population of UNCHARGED form (i.e. Deprotonated form) of the carboxyl groups would also be Higher. So, according to the equation proposed by the authors, it is clear that the representation of Protonation and Deprotonation is not correct, as shown in Figure 3. Therefore, Figure 3 must be modified, so as to have agreement with equation proposed for ‘koff’. In other words, the authors have defined ‘koff’ and ‘kon’equations correctly, but these equations do not accurately represent the Proton-bound form and the Deprotonated form, particularly for the free Carboxyl groups.

Overall, the authors are urged to provide clarification for the proper definition of ON-state and OFF-state, by taking into account the conditions that are shown below:

(i) for Free Carboxyl Groups (whose pKa values are closer to acidic pH):

Deprotonation = NEGATIVELY CHARGED Form = -COO for pH > ~ 2.5 or 3 and

Reprotonation = UNCHARGED Form = -COOH for pH < ~ 2

Therefore, Deprotonation = ON-state (Higher Voltage) and

Reprotonation = OFF-state (Lower Voltage)

(ii) for Free Amino Groups (whose pKa values are higher than acidic pH):

Deprotonation = UNCHARGED Form = -NH2 for pH > 10 (approx.) and

Reprotonation = POSITIVELY CHARGED Form = -N+H3 in the pH range: 2 – 10.

Therefore, Deprotonation = OFF-state (Lower Voltage) and

Reprotonation = ON-state (Higher Voltage)

Thus, the representation shown in Figure 3 is correct for Free Amino Groups and not for Free Carboxyl Groups. The authors need to make a note of this and do the necessary corrections.

Deprotonation is the transition from on-state to off-state, and reprotonation is the reverse transition. This has been clarified in the manuscript, with explicit mention of the effect these transitions have on the charge of the end-group of the molecule.

5. Digestion of proteins would give rise to many peptides. So, how is it ensured that only One Molecule of the Peptide binds to a chip? Is the chip designed in a manner that only One Molecule of the Peptide can be accommodated on its surface? It is established that chromatographic methods aid in decreasing the complexity of the mixture of peptides. So, before allowing the digested peptides to bind onto the chip surface, is the mixture of digested peptides subjected to any chromatographic method? In other words, are the peptides introduced to bind onto the chip surface, after any chromatographic method? Or, is the mixture of digested peptides directly introduced onto the chip surface for binding?

We clarify the protein sample treatment procedure (1) and surface functionalization (2) below

 (1) To obtain N-terminal peptides of proteins only, the protease cleavage of the protein sample is combined with an N-terminal peptide separation chromatography technique [a] such as for example COFRADIC [b]. Chromatographically separating the signature N-terminal peptides from the protease digested protein sample allows to significantly reduce the peptide sample complexity while still allowing identification of the original proteins. This technique is applied in mass spectroscopy-based proteomics[a]. Alternatively, to obtain N-terminal peptides the proteins can be denatured and bound covalently with their N-terminal to the sensor devices on the chip as described below, and subsequently cleaved by a protease. 

(2) In order to obtain approximately a single molecule on a single sensor device a Poissonian delivery technique can be used as is applied in the commercially available DNA sequencer of Pacific Biosciences [c]. Such technique leads to a random distribution of the amount of macromolecules per sensor limited by Poissonian statistics with a theoretical maximum single-molecule occupancy of 37%. A higher single molecule occupation can be obtained for an optimized delivery technique [d]. Molecules can be bound covalently to a surface for instance with Cu-free strain-promoted azide alkyne click chemistry (SPAAC).

[a] Principles of proteomics by RM Twyman 2014

[b] Gevaert K, Goethals M, Martens L, Van Damme J, Staes A, Thomas GR, Vandekerckhove J., “Exploring proteomes and analyzing protein processing by mass spectrometric identification of sorted N-terminal peptides.”, Nat Biotechnol. 2003 May;21(5):566-9

[c] Korlach J, Marks PJ, Cicero RL, Gray JJ, Murphy DL, Roitman DB, Pham TT, Otto GA, Foquet M, Turner SW, “Selective Aluminum Passivation for Targeted Immobilization of Single DNA Polymerase Molecules in Zero-Mode Waveguide Nanostructures.”, Proc. Natl. Acad. Sci. U. S. A. 2008, 105, 1176−1181.

[d] Thomas Plénat, Satoko Yoshizawa, and Dominique Fourmy, “DNA-Guided Delivery of Single Molecules into Zero-Mode Waveguides”, ACS Appl. Mater. Interfaces 2017, 9, 30561-30566

We have added these explanations in the manuscript. 

6. The authors need to mention the source, from where the pKa values have been taken to prepare Table 1.

A reference [Brown W, Foote C, Iverson B, Anslyn E (2010) Organic Chemistry] has been provided.

7. What do the authors mean by ‘clusters’? It is better to give a proper definition for ‘clusters’ and then describe about them. How a cluster is different from sequence?

We use “cluster” in the context of cluster analysis, i.e. a cluster is a grouping of amino acids such that amino acids in the same cluster are harder to distinguish from each other, than amino acids in distinct clusters. Clusters are written as a minimal sequence of the contained amino acids for the sake of brevity, this has been clarified in the manuscript.

8. Why the authors have not compared their simulation results with some experimental data that may be already available?

There is no experimental data available at this point, and as such there is no comparison possible between simulations and experiments.

Reviewer #3: In this manuscript, the authors propose a new single-molecule protein sequencing method. The scheme is based on measurements of the average time an amino acids spends in a protonation states using field effector transistor (FET). The concept is novel, which makes the manuscript suitable for publication in Plos One. But the feasibility of this method is not high, so the authors should answer several concerns for the publication.

1) The authors claim that a carbon nanotube FET can recognize the gate electric field generated by a single charge. But it is unlikely that the gate electric field generated by a single charge can give such a significant effect on the carbon nanotube channel in a real experiment. There are many other factors contributing to fluctuations in the electrical signal, such as electrical noise, thermal fluctuations, and electrolyte ion migration, which would result in larger effects. This will be much more extreme than the cases in the references (17-22) since the proposed device measures only a single charge. This concern should be addressed in the manuscript.

It is indeed the case that it remains to be proven that FET devices can attain the elementary charge signal to noise ratios that are required for deprotonation-protonation sequencing. We have made this more clear in the abstract:

“Our simulations provide target single elementary charge signal-to-noise ratios that sensor devices need to attain in order to make reprotonation-deprotonation protein sequencing possible. For instance, we calculate that an SNR of 10 is required for a 61.71% proteome recovery rate with 100 ms measurement time per amino acid. The simulations also provide required measurement times and suitable pH values.”

We have made it more clear as well in the ‘hardware’ section that the electrical detection of deprotonation/protonation of a single molecule remains to be proven: “Realizing a sensor that can monitor the protonation state of a single molecule with a sufficiently high signal-to-noise ratio is an outstanding challenge.” 

A particular published work which indicates that it is indeed possible to distinguish a single proton in a biomolecule with a carbon nanotube FET is that of Choi et al. This has been made more clear now in the ‘hardware’ section text:

“By recombinantly modifying the charge of two amino acid residues in the protein lysozyme, Choi et al. make protein variants differing in charge by elementary units N=-2,-1,0,1 and 2. Carbon nanotube FETs were able to distinguish these elementary charge differences, indicating that carbon nanotubes have the potential to distinguish protonation-deprotonation of amino acids.” 

2) Related to (1), a 2004 paper (ref 16) is cited where a single peptide was measured. Yet in this reference, a very simplified model peptide was used, where only one amino acid side chain group could dissociate. It is not obvious how signals could be resolved when multiple dissociating groups would present when analysing peptides derived from proteins. This concern should be addressed in the manuscript.

Ref 16 indeed uses a simple reference peptide. We have accounted for protonation and deprotonation of side chains of the peptide amino acids in the simulations. Peptide side chain impact is discussed in several parts of the text amongst which in the conclusion: “First, side chain reprotonation-deprotonation emits a secondary signal with a higher amplitude than the main signal. This leads to a significant decrease in accuracy with the identification method that we applied in this work. More sophisticated techniques could be developed to accurately discern the main signal from the side chain signals. As a matter of fact, the side chains could actually contribute to a more precise identification of the amino acids in the peptide, at a higher data processing cost.” 

In the methods section: 

“In the presence of side chains, the measured on-fraction can be affected. If the pH of the solution is significantly higher or lower than the pKa of the side chain, then the side chain is almost always in a fixed state, i.e. protonated or deprotonated, and this is not an issue. The only way to lock all side chains in this manner without also removing the main reprotonation-deprotonation signal, is by selecting a very acidic pH (<1) and measuring the reprotonation-deprotonation of the carboxyl group. Such a low pH is unlikely to be feasible in practice. However, since there are basic and acidic side chains, at any given pH, at least some of them are unlikely to change states in any given measurement at a fixed pH. An alternative approach is to effectively block the sidechain reprotonation-deprotonation signal by binding another molecule to it. In practice, these blocking molecules do not have 100% binding efficiency, and some side chain reprotonation-deprotonation signals will remain present in the measured signal.” 

3) Authors discuss the read error from shielding, which stems from the significant distance from the peptide end to the surface (effect related to the measuring device). However, one can expect that local intramolecular interaction (hydrogen bonding etc.) will also affect the pKa of particular amino acid residues. This error of pKa variance is not modelled or discussed.

Amino acid pKa can depend on neighboring amino acids. We discuss this in the section ‘simulation process’: “Different pKa values have been reported for the same amino acids [24]. In practice, the pKa of amino acids are site-specific, i.e. they depend on the location of the peptide in the protein [25]. Using a more advanced identification algorithm this can be taken into account by learning the pKa values based on known peptide databases.”

4) It would be challenging to make a sensor that is able to withstand repeatedly chemicals used in Edmann degradation or one where proteins would not get adsorbed non-specifically. Please discuss these issues.

Chip making materials such as carbon nanotubes and silicon oxide are chemically resistant. Protein adsorption can be avoided by making use of for instance polyethylene glycol (PEG) coatings or polyvinylphosphonic acid (PVPA) [d]. In work by Borgo and Havranek an enzyme Edmanase [e] is proposed to carry out Edman degradation without the harsh acid catalysis step which improves the compatibility with low adsorption detection surfaces. We have added this discussion to the manuscript. 

[d] Thomas Plénat, Satoko Yoshizawa, and Dominique Fourmy, “DNA-Guided Delivery of Single Molecules into Zero-Mode Waveguides”, ACS Appl. Mater. Interfaces 2017, 9, 30561-30566

[e] Borgo B, Havranek JJ Computer-aided design of a catalyst for Edman degradation utilizing substrate-assisted catalysis, Protein Science 2015 Vol. 24:571—579 571 

Minor comments

1) Figure 1 does not accurately describe how detection would be achieved. Please revise it.

The figure has been revised.

2) In page 3, the deprotonation scheme is incorrect for N-terminal amine: -CH3+->CH2 + H+. Carbon should be replaced with nitrogen.

This has been corrected in the manuscript.

3) pKa values of dissociating groups of amino acids are different among literature. The authors should comment on their choice.

A reference has been provided [Brown W, Foote C, Iverson B, Anslyn E (2010) Organic Chemistry]. There is no underlying reason for this particular choice.

---

## [Decision Letter · Decision Letter 1]

22 May 2020

PONE-D-19-32444R1

Computational assessment of the feasibility of protonation-based protein sequencing

PLOS ONE

Dear Dr. Miclotte,

Thank you for submitting your manuscript to PLOS ONE. After careful consideration, we feel that it has merit but does not fully meet PLOS ONE’s publication criteria as it currently stands. Therefore, we invite you to submit a revised version of the manuscript that addresses the points raised during the review process.

Two reviewers raised some new questions and helpful suggestions for the further improvement of the writing of the manuscript. The authors need to address these questions and revise the manuscript carefully. Particularly, the comments and suggestions by the reviewer 2 should be well considered in revising their manuscript.

We look forward to receiving your revised manuscript.

Kind regards,

Wenfei Li, Ph.D.

Academic Editor

PLOS ONE

Additional Editor Comments (if provided):

Two reviewers raised some new questions and helpful suggestions for the further improvement of the writing of the manuscript. The authors need to address these questions and revise the manuscript carefully. Particularly, the comments and suggestions by the reviewer 2 should be well considered in revising their manuscript.

Reviewers' comments:

Reviewer's Responses to Questions

**Comments to the Author**

1. If the authors have adequately addressed your comments raised in a previous round of review and you feel that this manuscript is now acceptable for publication, you may indicate that here to bypass the “Comments to the Author” section, enter your conflict of interest statement in the “Confidential to Editor” section, and submit your "Accept" recommendation.

Reviewer #2: (No Response)

Reviewer #3: All comments have been addressed

2. Is the manuscript technically sound, and do the data support the conclusions?

Reviewer #2: Partly

Reviewer #3: Yes

3. Has the statistical analysis been performed appropriately and rigorously? 

Reviewer #2: N/A

Reviewer #3: N/A

4. Have the authors made all data underlying the findings in their manuscript fully available?

Reviewer #2: Yes

Reviewer #3: Yes

5. Is the manuscript presented in an intelligible fashion and written in standard English?

Reviewer #2: Yes

Reviewer #3: Yes

6. Review Comments to the Author

Reviewer #2: Comments to the Authors for the manuscript: PONE-D-19-32444R1

1. The authors have now clarified the definition of ‘ON’ and ‘OFF’ with respect to the Carboxyl group, whereby they have ‘reiterated’ that ‘Deprotonated state (Negatively charged form)’ corresponds to ‘OFF-state’ and ‘Protonated state (Neutral form)’ corresponds to ‘ON-state’.

But, a doubt that still prevails is that how a ‘Neutral form’ (Protonated form: COOH) can give rise to a higher voltage than the ‘Deprotonated form (Carboxylate form: COO-)’? In other words, how can a Neutral form give a response to the electronic or electric circuit, since the Neutral form does NOT carry any Charge? However, the ‘Deprotonated (Negatively charged) form’ can be detected by the electronic or electric circuit, since it possesses a Charge, although it could show a signal in the negative axis of the output graph, i.e., negative voltage or negative potential. So, can the authors clarify this? If it would have been an experimental work, then this question could be answered in a more convincing manner. However, this is a computational work and not experimental. Therefore, it becomes even more important that such definitions need to be proposed lucidly and properly.

2. In page 14/17, in the section, “Underlying terminal reprotonation-deprotonation signal”: the equations for ‘ON-time’ and ‘OFF-time’ must be clarified. Currently, the authors have defined as:

(a) ON-time ~ Exp (koff), and

(b) OFF-time ~ Exp (kon)

This definition seems to give a wrong impression. As per this equation, when koff is higher, then ON-time is higher. Likewise, higher the value of kon, higher will be OFF-time. The authors need to clarify, as to, how is this possible? In other words, how ON-time can be ‘directly proportional’ to koff and how is it possible for the OFF-time to be ‘directly proportional’ to kon?

When koff is a higher value, then the system (i.e., the molecule) would tend to be in OFF-state (i.e., Deprotonated form), MORE Frequently than in the ON-state. In other words, ‘OFF’ state would be detected or observed MORE frequently than the ‘ON’ state. So, the question is: “During a particular time period of measurement, if the ‘OFF-state’ would be detected at a higher frequency than ON-state, then how ‘ON-time’ can be directly proportional to koff? When ‘OFF-time’ would be observed at a higher frequency than ON-state, it means that the Total OFF-time (also the Average OFF-time) would be greater than the Total ON-time (also the Average ON-time), during a particular time period of measurement, i.e.,

Total OFF-time > Total ON-time, which implies, Average OFF-time > Average ON-time

Therefore, it is not clear, as to, on what basis or rationale, the authors have proposed these two definitions? The authors have mentioned, “the associative and dissociative events occur continuously, independently and at a constant average rate of koff and kon. But, what is the relationship or connection between this statement, i.e., “association and dissociation events occur continuously, independently, at a constant average rate of koff and kon”; and the equations defined for ON-time and OFF-time?

3. The method proposed in this work is appropriate for ‘pKa measurement’, rather than for protein sequencing. Further, the proposed strategy may also be applicable to determine ‘isoelectric point’ of protein. However, in the context of protein sequencing, this proposed method of measuring reprotonation and deprotonation events, can become meaningful and useful, only when this method is combined with proteolysis (by using endopeptidase) or by combining with Edman’s method. Another essential requirement is a proper reference database containing protein sequences. So, without involvement of proteolysis or Edman’s method and in the absence of a reference database, this method can only be useful to measure pKa or isoelectric point of proteins. Therefore, what is the need to mention ‘protein sequencing’ in the title of this manuscript? Instead, the authors may change the title, perhaps by emphasizing ‘pKa measurement’ of single molecule of protein. The authors could also add a phrase in the title by mentioning: ‘application to protein sequencing’. So, this manuscript may be titled as “pKa measurement of protein using (carbon nanotube) FET for protein sequencing”. Or it may be titled as, “Single molecule pKa measurement (by carbon nanotube FET) for protein sequencing”. The authors can decide, as to, whether to mention about FET or not, in the title?

Therefore, accordingly, in the list of keywords, ‘pKa measurment’ or ‘pKa’ can be included. Even ‘Isoelecric point (pI)’ can be mentioned in the list of keywords.

Reviewer #3: The authors have sufficiently addressed my concerns and have improved their manuscript. I recommend the publication of this manuscript.

A minor comment:

The reference number [27] in the ‘Hardware’ section on page 5 seems to be incorrect.

“By recombinantly modifying two amino acid residues in the protein lysozyme, Choi et al. produced protein variants that differ in charge by elementary units N=-2, -1 ,0, 1, and 2. Carbon nanotube FETs were able to distinguish these elementary charge differences, indicating that carbon nanotubes have the potential to monitor the protonation-deprotonation of amino acids [27].”

The reference number would be [24].

7. PLOS authors have the option to publish the peer review history of their article (what does this mean?). If published, this will include your full peer review and any attached files.

Reviewer #2: Yes: Varatharajan Sabareesh

Reviewer #3: No

---

## [Author Response · Author response to Decision Letter 1]

3 Jul 2020

Reviewer #2

1. The authors have now clarified the definition of ‘ON’ and ‘OFF’ with respect to the Carboxyl group, whereby they have ‘reiterated’ that ‘Deprotonated state (Negatively charged form)’ corresponds to ‘OFF-state’ and ‘Protonated state (Neutral form)’ corresponds to ‘ON-state’.

But, a doubt that still prevails is that how a ‘Neutral form’ (Protonated form: COOH) can give rise to a higher voltage than the ‘Deprotonated form (Carboxylate form: COO-)’? In other words, how can a Neutral form give a response to the electronic or electric circuit, since the Neutral form does NOT carry any Charge? However, the ‘Deprotonated (Negatively charged) form’ can be detected by the electronic or electric circuit, since it possesses a Charge, although it could show a signal in the negative axis of the output graph, i.e., negative voltage or negative potential. So, can the authors clarify this? If it would have been an experimental work, then this question could be answered in a more convincing manner. However, this is a computational work and not experimental. Therefore, it becomes even more important that such definitions need to be proposed lucidly and properly.

Reply: We consider the change in voltage, and this is proportional to the change in charge at the proptonation sites of the peptide. As such the neutral voltage due to the neutral charge is considered to be higher than the negative voltage induced by the negatively charged carboxyl group. We now added explicit mention of this in the results section and in the methods section.

2. In page 14/17, in the section, “Underlying terminal reprotonation-deprotonation signal”: the equations for ‘ON-time’ and ‘OFF-time’ must be clarified. Currently, the authors have defined as:

(a) ON-time ~ Exp (koff), and

(b) OFF-time ~ Exp (kon)

This definition seems to give a wrong impression. As per this equation, when koff is higher, then ON-time is higher. Likewise, higher the value of kon, higher will be OFF-time. The authors need to clarify, as to, how is this possible? In other words, how ON-time can be ‘directly proportional’ to koff and how is it possible for the OFF-time to be ‘directly proportional’ to kon?

When koff is a higher value, then the system (i.e., the molecule) would tend to be in OFF-state (i.e., Deprotonated form), MORE Frequently than in the ON-state. In other words, ‘OFF’ state would be detected or observed MORE frequently than the ‘ON’ state. So, the question is:

“During a particular time period of measurement, if the ‘OFF-state’ would be detected at a higher frequency than ON-state, then how ‘ON-time’ can be directly proportional to koff? When ‘OFF-time’ would be observed at a higher frequency than ON-state, it means that the Total OFF-time (also the Average OFF-time) would be greater than the Total ON-time (also the Average ON-time), during a particular time period of measurement, i.e.,

Total OFF-time > Total ON-time, which implies, Average OFF-time > Average ON-time

Therefore, it is not clear, as to, on what basis or rationale, the authors have proposed these two definitions? The authors have mentioned, “the associative and dissociative events occur continuously, independently and at a constant average rate of koff and kon. But, what is the relationship or connection between this statement, i.e., “association and dissociation events occur continuously, independently, at a constant average rate of koff and kon”; and the equations defined for ON-time and OFF-time?

Reply: We made a change to the manuscript to clarify the meaning of Exp(k). The on-time is distributed according to the probability density function k_off * e^(-k_off * x), and as such, if k_off increases, the average on-time decreases, and similarly for k_on and the off-time.

3. The method proposed in this work is appropriate for ‘pKa measurement’, rather than for protein sequencing. Further, the proposed strategy may also be applicable to determine ‘isoelectric point’ of protein. However, in the context of protein sequencing, this proposed method of measuring reprotonation and deprotonation events, can become meaningful and useful, only when this method is combined with proteolysis (by using endopeptidase) or by combining with Edman’s method. Another essential requirement is a proper reference database containing protein sequences. So, without involvement of proteolysis or Edman’s method and in the absence of a reference database, this method can only be useful to measure pKa or isoelectric point of proteins. Therefore, what is the need to mention ‘protein sequencing’ in the title of this manuscript? Instead, the authors may change the title, perhaps by emphasizing ‘pKa measurement’ of single molecule of protein. The authors could also add a phrase in the title by mentioning: ‘application to protein sequencing’. So, this manuscript may be titled as “pKa measurement of protein using (carbon nanotube) FET for protein sequencing”. Or it may be titled as, “Single molecule pKa measurement (by carbon nanotube FET) for protein sequencing”. The authors can decide, as to, whether to mention about FET or not, in the title?

Therefore, accordingly, in the list of keywords, ‘pKa measurment’ or ‘pKa’ can be included. Even ‘Isoelecric point (pI)’ can be mentioned in the list of keywords.

Reply: While this method could be applied in a more general sense for pKa measurement, this manuscript is written from the perspective of protein sequencing. As such, we feel that the current title covers the content well.

Reviewer #3:

The authors have sufficiently addressed my concerns and have improved their manuscript. I recommend the publication of this manuscript.

A minor comment:

The reference number [27] in the ‘Hardware’ section on page 5 seems to be incorrect.

“By recombinantly modifying two amino acid residues in the protein lysozyme, Choi et al. produced protein variants that differ in charge by elementary units N=-2, -1 ,0, 1, and 2. Carbon nanotube FETs were able to distinguish these elementary charge differences, indicating that carbon nanotubes have the potential to monitor the protonation-deprotonation of amino acids [27].”

The reference number would be [24].

Reply: The reference has been corrected.

---

## [Decision Letter · Decision Letter 2]

18 Aug 2020

PONE-D-19-32444R2

Computational assessment of the feasibility of protonation-based protein sequencing

PLOS ONE

Dear Dr. Miclotte,

Thank you for submitting your manuscript to PLOS ONE. After careful consideration, we feel that it has merit but does not fully meet PLOS ONE’s publication criteria as it currently stands. Therefore, we invite you to submit a revised version of the manuscript that addresses the points raised during the review process.

Additional Editor Comments (if provided):

The authors need to make a minor revision to improve the readability of the manuscript as suggested by our reviewer before I can recommend for publication.

We look forward to receiving your revised manuscript.

Kind regards,

Wenfei Li, Ph.D.

Academic Editor

PLOS ONE

Reviewers' comments:

Reviewer's Responses to Questions

**Comments to the Author**

1. If the authors have adequately addressed your comments raised in a previous round of review and you feel that this manuscript is now acceptable for publication, you may indicate that here to bypass the “Comments to the Author” section, enter your conflict of interest statement in the “Confidential to Editor” section, and submit your "Accept" recommendation.

Reviewer #2: (No Response)

2. Is the manuscript technically sound, and do the data support the conclusions?

Reviewer #2: Partly

3. Has the statistical analysis been performed appropriately and rigorously? 

Reviewer #2: N/A

4. Have the authors made all data underlying the findings in their manuscript fully available?

Reviewer #2: Yes

5. Is the manuscript presented in an intelligible fashion and written in standard English?

Reviewer #2: Yes

6. Review Comments to the Author

Reviewer #2: Comments to the Authors for the manuscript: PONE-D-19-32444R2

1. In places, where the Protonated Form of Amine group is shown, the POSITIVE CHARGE (+) MUST be positioned NEAR the Nitrogen Atom, i.e., as a superscript to Nitrogen atom, on it's left-hand side, as shown here: - +NH3.

Currently, it is shown as ‘superscript’ of the Hydrogen Atom, which is NOT Correct. So, the authors MUST make this correction, before the manuscript is taken up for publication.

2. In page 3/17, in the section ‘Protonation-based protein sequencing’, second paragraph, line no. 8: ‘mass spectroscopy’ must be corrected to ‘mass spectrometry’. So, it must read as ‘mass spectrometry-based proteomics’.

3. In page 6/17, second paragraph, line no. 7: …..the voltage difference between the on-state and off-state., can be mentioned within inverted commas, viz., “the voltage difference between the on-state and off-state.” In doing so, this point might get more emphasis, thereby enabling better understanding for the readers.

4. In page 14/17, in the section, “Underlying terminal reprotonation-deprotonation signal”: clarification for the equations for ‘ON-time’ and ‘OFF-time’ was requested earlier.

(a) ON-time ~ Exp (koff), and

(b) OFF-time ~ Exp (kon)

In response to this query, the authors have now added a sentence and included a mathematical function, f(x) = k e(-kx), which was not mentioned in the earlier two versions of this manuscript. By including this mathematical function, there seems to be a better clarity and understanding in the definitions about ON-time and OFF-time. However, the authors are urged to provide some detailed explanation for the ‘k’ in this function, f(x) = k e(-kx).

(a) What is the ‘k’ that is appearing as co-efficient to the exponential function?

&

(b) What is the ‘k’ that is placed within the exponential function along with ‘x’?

Are these two ‘k’ same or different? If these two ‘k’ are same, does this ‘k’ refer to koff and kon in the respective equations for ON-time and OFF-time? If these two ‘k’ are different, then the authors must use another symbol for the co-efficient of this exponential function. And it seems that the ‘k’ which is placed within the exponential function along with ‘x’, corresponds to koff and kon for the respective equations of ON-time and OFF-time. Kindly clarify.

7. PLOS authors have the option to publish the peer review history of their article (what does this mean?). If published, this will include your full peer review and any attached files.

Reviewer #2: No

---

## [Author Response · Author response to Decision Letter 2]

18 Aug 2020

1. In places, where the Protonated Form of Amine group is shown, the POSITIVE CHARGE (+) MUST be positioned NEAR the Nitrogen Atom, i.e., as superscript to Nitrogen atom, on it’s left-hand side, as shown here: - +NH 3 . Currently, it is shown as ‘superscript’ of the Hydrogen Atom, which is NOT Correct. So, the authors MUST make this correction, before the manuscript is taken up for publication.

 - Changed as requested.

2. In page 3/17, in the section ‘Protonation-based protein sequencing’, second paragraph, line no. 8: ‘mass spectroscopy’ must be corrected to ‘mass spectrometry’. So, it must read as ‘mass spectrometry-based proteomics’.

 - Changed as requested.

3. In page 6/17, second paragraph, line no. 7: .....the voltage difference between the on-state and off- state., can be mentioned within inverted commas, viz., “the voltage difference between the on-state and off-state.” In doing so, this point might get more emphasis, thereby enabling better understanding for the readers.

 - Changed as requested.

4. In page 14/17, in the section, “Underlying terminal reprotonation-deprotonation signal”: clarification for the equations for ‘ON-time’ and ‘OFF-time’ was requested earlier.

(a) ON-time ~ Exp (k off ), and

(b) OFF-time ~ Exp (k on )

In response to this query, the authors have now added a sentence and included a mathematical function, f(x) = ke^{-kx} , which was not mentioned in the earlier two versions of this manuscript. By including this mathematical function, there seems to be a better clarity and understanding in the definitions about ON-time and OFF-time. However, the authors are urged to provide some detailed explanation for the ‘k’ inthis function, f(x) = ke^{-kx} .

(a) What is the ‘k’ that is appearing as co-efficient to the exponential function?

(b) What is the ‘k’ that is placed within the exponential function along with ‘x’?

Are these two ‘k’ same or different? If these two ‘k’ are same, does this ‘k’ refer to k off and k on in the respective equations for ON-time and OFF-time? If these two ‘k’ are different, then the authors mustuse another symbol for the co-efficient of this exponential function. And it seems that the ‘k’ which is placed within the exponential function along with ‘x’, corresponds to k off and k on for the respective equations of ON-time and OFF-time. Kindly clarify.

 - Both occurrences of ‘k’ are the same and refer to k off and k on in the respective equations for ON-time and OFF-time. We clarified the meaning of ‘k’. The sentence now reads:

 - "Exp(k) is the exponential distribution with rate parameter k and probability density function f(x) = ke^{-kx}. The on-time distribution has rate parameter k = k_{off}, while the off-time distribution has rate parameter k = k_{on}."

---

## [Editor Report · Decision Letter 3]

21 Aug 2020

Computational assessment of the feasibility of protonation-based protein sequencing

PONE-D-19-32444R3

Dear Dr. Miclotte,

We’re pleased to inform you that your manuscript has been judged scientifically suitable for publication and will be formally accepted for publication once it meets all outstanding technical requirements.

Kind regards,

Wenfei Li, Ph.D.

Academic Editor

PLOS ONE
---

## [Editor Report · Acceptance letter]

31 Aug 2020

PONE-D-19-32444R3 

Computational assessment of the feasibility of protonation-based protein sequencing 

Dear Dr. Miclotte:

I'm pleased to inform you that your manuscript has been deemed suitable for publication in PLOS ONE. Congratulations! Your manuscript is now with our production department. 

Kind regards, 

on behalf of

Dr. Wenfei Li 

Academic Editor

PLOS ONE